# ADVERSARIAL ROBUSTNESS AGAINST MULTIPLE $l_p$-THREAT MODELS AT THE PRICE OF ONE AND HOW TO QUICKLY FINE-TUNE ROBUST MODELS TO ANOTHER THREAT MODEL

## ABSTRACT

Adversarial training (AT) in order to achieve adversarial robustness wrt single $l_p$-threat models has been discussed extensively. However, for safety-critical systems adversarial robustness should be achieved wrt all $l_p$-threat models simultaneously. In this paper we develop a simple and efficient training scheme to achieve adversarial robustness against the union of $l_p$-threat models. Our novel E-AT scheme is based on geometric considerations of the different $l_p$-balls and costs as much as normal adversarial training against a single $l_p$-threat model. Moreover, we show that using our E-AT scheme one can fine-tune with just 3 epochs *any* $l_p$-robust model (for $p \in \{1, 2, \infty\}$) and achieve multiple norm adversarial robustness. In this way we boost the state-of-the-art for multiple-norm robustness to more than $51\%$ on CIFAR-10 and report up to our knowledge the first ImageNet models with multiple norm robustness. Moreover, we study the general transfer of adversarial robustness between different threat models and in this way boost the previous SOTA $l_1$-robustness on CIFAR-10 by almost $10\%$.

## 1 INTRODUCTION

The problem of adversarial examples, that is small adversarial perturbations of the input (Szegedy et al., 2014; Kurakin et al., 2017) changing the decision of a classifier, is a serious obstacle for the use of machine learning in safety-critical systems. Many adversarial defenses have been proposed but most of them could be broken either by stronger attacks with a higher computational budget (Carlini & Wagner, 2017; Athalye et al., 2018; Mosbach et al., 2018) or using adaptive attacks (Tramèr et al., 2020). Apart from provable adversarial defenses which are however still restricted to rather simple CNNs (Wong et al., 2018; Gowal et al., 2018), the only successful technique so far remains adversarial training (Madry et al., 2018) and its improvements (Zhang et al., 2019; Carmon et al., 2019; Wu et al., 2020; 2021). We refer to Gowal et al. (2020) for a recent overview of "tricks of the trade" for improving adversarial training yielding the currently most robust models for $l_2$ and $l_\infty$ on CIFAR-10. While the community initially focused on adversarial examples for $l_\infty$-perturbations, there has been recently more interest in other $l_p$-attacks, such as $l_1$ and $l_2$, or perceptual threat models (Stutz et al., 2019; Wong & Kolter, 2021; Laidlaw et al., 2021). It is well known that robustness in one $l_p$-ball does not necessarily generalize to some other $l_q$-ball for $p \neq q$ (Kang et al., 2019a; Tramèr & Boneh, 2019). However, in safety-critical systems we need robustness against all $l_p$-norms simultaneously which has triggered recent extensions of adversarial training for multiple $l_p$-norms (Tramèr & Boneh, 2019; Maini et al., 2020) and provable defenses for all $l_p$ with $p \geq 1$ (Croce & Hein, 2020b).

In this paper we show that, using the geometry of the $l_p$-balls, the computationally expensive multiple norm training procedures of Tramèr & Boneh (2019); Maini et al. (2020), which cost up to three times as much as normal adversarial training, can be replaced by a very effective and simple form of adaptively alternating between the two extreme norms, namely $l_1$ and $l_\infty$. This simple scheme achieves similar robustness for the union of the threat models as more costly previous approaches. Moreover, we show that already 3 epochs of fine-tuning with our extreme norms adversarial training (E-AT) are enough to turn any $l_p$-robust model for $p \in \{1, 2, \infty\}$ into a model which is robust

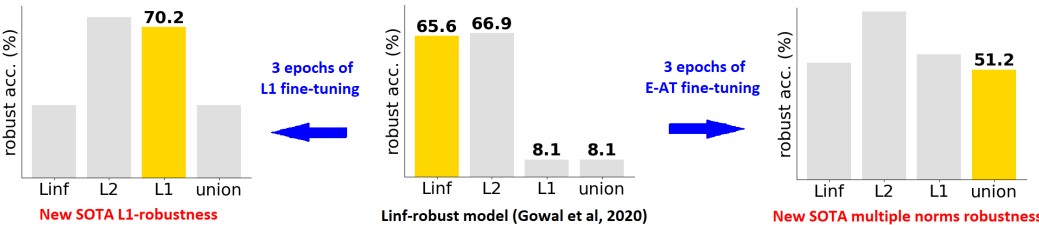

Figure 1: We fine-tune for 3 epochs the WideResNet-70-16 on CIFAR-10 from Gowal et al. (2020) with highest $l_\infty$-robustness to be either robust wrt $l_1$ (left) or with our E-AT to be robust wrt to the union of the $l_\infty$-, $l_2$-, and $l_1$-threat models (right). We achieve state-of-the-art results in both threat models. The plots show the robust accuracy in the individual threat models and in their union for the initial $l_\infty$-robust classifier (middle) and the fine-tuned ones, with the target threat model highlighted.

against all $l_p$-threat models for $p \in \{1, 2, \infty\}$, even if the original classifier was completely non-robust against one of the threat models. We apply the proposed method to many existing models originally trained to be robust in a single norm: fine-tuning the currently most robust network in the $l_\infty$-threat model from Gowal et al. (2020) for CIFAR-10 we improve the current state-of-the-art performance for multiple norm robustness by more than $6\%$ (robustness over the union of $l_1$, $l_2$ and $l_\infty$-balls). Moreover, we employ our E-AT fine-tuning scheme with just a single epoch to yield the first ImageNet model which is robust against multiple attacks at the same time. Finally, we show that fine-tuning with just *3 epochs* for CIFAR-10 and *one epoch* for ImageNet is sufficient to transfer robustness from one threat model to another one which very quickly yields baselines for all threat models. In this way we achieve SOTA $l_1$-robustness on CIFAR-10 by fine-tuning the most robust $l_\infty$-model from Gowal et al. (2020) to become $l_1$-robust, and get an ImageNet model robust wrt $l_1$. These results are quite striking as the original classifiers show no or only very little $l_1$-robustness. Fig. 1 summarizes the results of robust fine-tuning on CIFAR-10 (see Table 2 and 4 for details).

## 2 RELATED WORK

**Adversarial training:** In image classification adversarial examples were first described by Szegedy et al. (2014) even though earlier discussion of adversarial examples in email spam classification can be found in Dalvi et al. (2004); Lowd & Meek (2005). An early approach of adversarial training was the Fast Gradient Sign Method (FGSM) (Goodfellow et al., 2015), which then was extended to a multi-step attack in Kurakin et al. (2017). Nowadays, adversarial training as formulated in Madry et al. (2018) as a min-max optimization problem has been one of the few adversarial defenses which could not be broken by stronger attacks (Athalye et al., 2018). Other types of defenses use more sophisticated techniques, typically preventing the direct optimization of the attack. However, adaptive attacks specifically designed for these defenses have often shown that these alternative techniques are non-robust or much less robust than claimed (Tramèr et al., 2020). Thus adversarial training remains the only general method ensuring adversarial robustness across architectures and datasets. Recent improvements have been achieved by using different objectives (Zhang et al., 2019), unlabeled data (Carmon et al., 2019), adversarial weights perturbations (Wu et al., 2020) and wider networks Wu et al. (2021). In Gowal et al. (2020) several recent variants were systematically explored and for a very large architecture, a WideResNet-70-16, they obtain the most robust models for $l_\infty$ (radius $\frac{8}{255}$) and $l_2$ (radius $0.5$) for CIFAR-10, which we use for our fine-tuning experiments.

**Multiple norm robustness:** It was early on discovered that adversarial robustness against a specific $l_p$ threat model does typically not transfer to $l_q$ threat models for $p \neq q$ (see Kang et al. (2019a); Tramèr & Boneh (2019) for extensive studies). On the other hand to achieve really reliable machine learning models $l_p$-robustness wrt all $p$ is necessary. The first approach to such a generally robust model was done in Schott et al. (2019) which uses multiple variational auto-encoders for an analysis by synthesis (ABS) architecture. While this model is restricted to MNIST, it is robust against $l_0$, $l_2$ and $l_\infty$-attacks. However, it has been recently shown that with a stronger black-box attack (Croce et al., 2020b) the $l_0$-robustness is significantly lower than originally claimed. Tramèr & Boneh (2019); Maini et al. (2020); Madaan et al. (2021) use variants of adversarial training to

achieve robustness in multiple norms. Since these are the most similar methods to ours, we present them in detail below. Madaan et al. (2021) additionally proposes a meta-learning approach where one learns optimal noise to augment the samples and uses consistency regularization to enforce similar predictions on clean, augmented and adversarial samples. Finally, Stutz et al. (2020) combine adversarial training with a reject option for the classifier by down-weighting the confidence of adversarial samples. Their models generalize to other threat models but the comparison to normal adversarially trained models is difficult as their model is non-robust without the reject option.

**Provable robustness:** In the area of provably robust defenses, Croce & Hein (2020b) motivated a regularization approach based on the geometry of the $l_p$-balls which enforces multiple-norm robustness during training which allows then to derive provable guarantees for multiple-norm robustness in contrast to the empirical evaluation of adversarial training. However, their approach works only for small network architectures and relatively small radii of the $l_p$-balls.

**Fine-tuning of robust models:** Fine-tuning of an existing neural network is a commonly used technique in deep learning (Goodfellow et al., 2016) to quickly adapt an existing model to a different objective e.g. for language models (Howard & Ruder, 2018). More recently, it has been shown in the area of adversarial robustness that fine-tuning of pre-trained models, possibly using self-supervision, yields better adversarial robustness (Hendrycks et al., 2019; Chen et al., 2020; Xu & Yang, 2020). In Jeddi et al. (2020) it is shown that fine-tuning of non-robust models with 10 epochs can yield robust models with the caveat that their robustness evaluation is done using only a single run of PGD with 20 steps. We are not aware of any prior work discussing fine-tuning to transform an existing robust model wrt a single $l_p$ into a robust model wrt multiple threat models or wrt another $l_q$-threat model. In particular, we show that one can fine-tune an $l_\infty$-robust model so that it becomes $l_1$-robust even though the original model has not been $l_1$-robust at all.

**Evaluation of adversarial robustness:** Many white-box attacks for $l_\infty$ (Madry et al., 2018; Gowal et al., 2019), $l_2$ (Madry et al., 2018; Carlini & Wagner, 2017) and $l_1$ (Chen et al., 2018; Modas et al., 2019; Rony et al., 2021) have been proposed as well as several black box attacks (Brendel et al., 2018; Liu et al., 2019; Cheng et al., 2019; Al-Dujaili & O'Reilly, 2020; Meunier et al., 2019; Zhao et al., 2019; Andriushchenko et al., 2020) for different threat models. It has recently been shown that AutoAttack (Croce & Hein, 2020c), a parameter-free ensemble of the white-box attacks APGD for the cross-entropy and DLR-loss, FAB-attack (Croce & Hein, 2020a) and the black-box Square-attack (Andriushchenko et al., 2020) is reliably evaluating adversarial robustness for $l_2$ and $l_\infty$. AutoAttack has recently been extended to $l_1$ (Croce & Hein, 2021) outperforming all existing state-of-the art attacks for $l_1$. Croce & Hein (2020c; 2021) showed that on models defended with adversarial training the two versions of APGD (with budget as in AutoAttack) already give an accurate robustness evaluation. As we have to evaluate very large models always for three threat models, we use those as a strong standard attack in our experiments.

## 2.1 Adversarial Training for the union of $l_1$-, $l_2$- and $l_\infty$-balls

Let us denote by $f_\theta : \mathbb{R}^d \to \mathbb{R}^K$ the classifier parameterized by $\theta \in \mathbb{R}^n$, with input $x \in \mathbb{R}^d$ and $f_\theta(x) \in \mathbb{R}^K$ where $K$ is the number of classes of the task. Let further $\mathcal{D} = \{(x_i, y_i)\}_i$ be the training set, with $y_i$ the correct label of $x_i$, and $\mathcal{L} : \mathbb{R}^K \times \mathbb{R}^K \to \mathbb{R}$ a given loss function. The aim is to enforce adversarial robustness in all multiple $l_p$-balls simultaneously, i.e., defining $B_p(\epsilon_p) = \{x \in \mathbb{R}^d : \|x\|_p \leq \epsilon_p\}$, the threat model is the union of the individual $l_p$-balls

$$\Delta = B_1(\epsilon_1) \cup B_2(\epsilon) \cup B_\infty(\epsilon_\infty),$$

which is a non convex set, since in practice the $\epsilon_p$ are chosen such that no $l_p$-ball contains any of the others. In adversarial training the worst case loss for input perturbation in the threat model, $\max_{\delta \in \Delta} \mathcal{L}(f_\theta(x_i + \delta), y_i)$, is minimized. Efficiently maximizing the loss $\mathcal{L}$ in the union of threat models is non-trivial and different approaches to extend adversarial training to this setting have been proposed. They basically differ in the way how the inner maximization problem is tackled.

**MAX:** Tramèr & Boneh (2019) suggest to run the three attacks for each $B_p(\epsilon_p)$ for $p \in \{1, 2, \infty\}$ independently and then use the one which realizes the highest loss, that is

$$\max_{\delta \in \Delta} \mathcal{L}(f_\theta(x_i + \delta), y_i) = \max_{p \in \{1, 2, \infty\}} \max_{\delta \in B_p(\epsilon_p)} \mathcal{L}(f_\theta(x_i + \delta), y_i).$$

This training optimizes directly the worst case in the union but comes at the price of being nearly three times as expensive as normal adversarial training wrt a single $l_p$-ball.

**AVG:** In their AVG alternative Tramèr & Boneh (2019) suggest to run the three attacks for each $B_p(\epsilon_p)$ for $p \in \{1, 2, \infty\}$ independently but replace the inner maximization problem with

$$\sum_{p \in \{1, 2, \infty\}} \max_{\delta \in B_p(\epsilon_p)} \mathcal{L}(f_\theta(x_i + \delta), y_i),$$

and thus one basically averages the updates of all $l_p$-balls with the motivation of not "wasting" the computed attacks, in particular when the attained loss values are rather similar and thus the max is ambiguous. Again this costs three times as much as normal adversarial training.

**MSD:** Maini et al. (2020) argue that the correct way to maximize the loss in the union is to test during the PGD attacks all the three steepest ascent updates corresponding to the three norms (sign of the gradient for $l_\infty$, normalized gradient for $l_2$ and a smoothed $l_1$-step by using the top-$k$ components in magnitude of the gradient) and then take the step which yields the highest loss. Maini et al. (2020) report that MSD outperforms both AVG and MAX, also in terms of a more stable training. As all three updates (forward passes) are tested but only one backward pass is needed (gradient is the same) this costs roughly two times as much as normal adversarial training.

**SAT:** Madaan et al. (2021) introduce Stochastic Adversarial Training (SAT) which randomly samples $p \in \{1, 2, \infty\}$ for each batch and performs PGD only for the corresponding $l_p$-norm. While SAT has the same cost as standard adversarial training, Madaan et al. (2021) report that it does not perform very well.

## 3 FAST MULTIPLE-NORM ROBUSTNESS VIA EXTREME NORMS ADVERSARIAL TRAINING AND FINE-TUNING

All previous methods assume that for achieving robustness to multiple norms each of the threat models has to be used at training time. In the following we first present an argument, using recent results from Croce & Hein (2020b), showing that this is not the case. Based on this analysis we introduce our extreme norms adversarial training (E-AT) which achieves multiple norm robustness at the same price as training for a single norm threat model. Finally, we show that only a few epochs of fine-tuning with E-AT turns a model robust wrt a single norm into one which has competitive robustness wrt multiple norms.

### 3.1 GEOMETRY OF THE UNION OF $l_p$-BALLS AND THEIR CONVEX HULL

The main insight we use for E-AT is that a linear classifier which is robust in both an $l_1$- and an $l_\infty$-ball is also robust wrt the largest $l_p$-ball for $1 \leq p \leq \infty$ which fits into the convex hull of the union of the $l_1$- and $l_\infty$-ball. This ball is significantly larger than the largest $l_p$-ball contained into the union of the $l_1$- and $l_\infty$-ball (see Fig. 2). Thus it is sufficient to be robust wrt the two "extreme" norms $l_1$ and $l_\infty$ to ensure robustness. While this is exact for affine classifiers, we conjecture that for neural networks this will at least hold approximately true (note that typical ReLU-networks yield piecewise affine classifiers (Arora et al., 2018)) and for the model it is the most efficient way in terms of capacity to be $l_1$- and $l_\infty$-robust.

We now state the main results from Croce & Hein (2020b) which are the basis for our E-AT. We work in the non-trivial setting, where the balls are not included in each other, that is $B_1(\epsilon_1) \not\subseteq B_\infty(\epsilon_\infty)$ and $B_\infty(\epsilon_\infty) \not\subseteq B_1(\epsilon_1)$ as otherwise the problem of enforcing multiple norms (including $l_1$ or $l_\infty$) robustness boils down again to single norm robustness. In order to be in this non-trivial setting it has to hold $\epsilon_1 \in (\epsilon_\infty, d\epsilon_\infty)$. For $\epsilon_\infty = \frac{8}{255}$ and dimension $d = 3072$ as in CIFAR-10 this yields an upper bound $\epsilon_1 \leq 96.38$ which is far higher than what has been used for $l_1$-threat models (we use $\epsilon_1 = 12$ for CIFAR-10). We denote by $U_{1,\infty}(\epsilon_1, \epsilon_\infty) = B_1(\epsilon_1) \cup B_\infty(\epsilon_\infty)$ the union of the $l_1$- and $l_\infty$-balls. One can then ask the obvious question: how much $l_p$-robustness do I get for $1 < p < \infty$ from a classifier robust in $U_{1,\infty}(\epsilon_1, \epsilon_\infty)$? This question is answered in the following result:

**Proposition 3.1 (Croce & Hein (2020b))** *If $d \geq 2$ and $\epsilon_1 \in (\epsilon_\infty, d\epsilon_\infty)$, then*

$$\min_{x \in \mathbb{R}^d \setminus U_{1,\infty}(\epsilon_1, \epsilon_\infty)} \|x\|_p = \left( \epsilon_\infty^p + \frac{(\epsilon_1 - \epsilon_\infty)^p}{(d-1)^{p-1}} \right)^{\frac{1}{p}}. \tag{1}$$

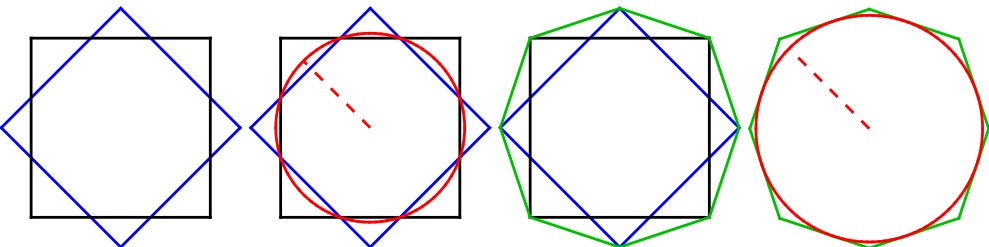

Figure 2: Visualization of the $l_2$-ball contained in the union resp. the convex hull of the union of $l_1$- and $l_\infty$-balls in $\mathbb{R}^2$. **First:** co-centric $l_1$-ball (blue) and $l_\infty$-ball (black). **Second:** in red the largest $l_2$-ball contained in the union of $l_1$- and $l_\infty$-ball. **Third:** in green the convex hull of the union of the $l_1$- and $l_\infty$-ball. **Fourth:** the largest $l_2$-ball (red) contained in the convex hull. The $l_2$-ball contained in the convex hull is significantly larger than that in the union of $l_1$- and $l_\infty$-ball.

Thus a classifier which is robust for the union $U_{1,\infty}(\epsilon_1, \epsilon_\infty)$ has automatically a non-trivial robustness for all intermediate $l_p$-norms. For us the case $p = 2$ is most interesting which given that $\epsilon_1 \gg \epsilon_\infty$ can be tightly upper bounded as

$$\epsilon_2 := \min_{x \in \mathbb{R}^d \setminus U_{1,\infty}(\epsilon_1, \epsilon_\infty)} \|x\|_2 \leq \sqrt{\epsilon_\infty^2 + \frac{\epsilon_1^2}{d-1}}. \tag{2}$$

For the case of CIFAR-10 where $d = 3072$ and $\epsilon_1 = 12$ and $\epsilon_\infty = \frac{8}{255}$ as chosen in Maini et al. (2020) one gets $\epsilon_2 \leq 0.2188$. As the radius of the $l_2$-threat model in Maini et al. (2020) is chosen as $\epsilon_2 = 0.5$, robustness in the union alone would not be sufficient to achieve the desired robustness wrt $l_p$ for $p \in \{1, 2, \infty\}$. But the above point of view is the worst case. In the following we see that if we consider affine classifiers then a guarantee for $B_1(\epsilon_1)$ *and* $B_\infty(\epsilon_\infty)$ implies a guarantee with respect to the convex hull $C$ of their union $B_1(\epsilon_1) \cup B_\infty(\epsilon_\infty)$ as an affine classifier generates a half-space and thus only the extreme points of $B_1(\epsilon_1)$ resp. $B_\infty(\epsilon_\infty)$ matter (see Figure 2 for illustrations of $B_1$, $B_\infty$, their union and their convex hull).

**Theorem 3.1 (Croce & Hein (2020b))** *Let $C$ be the convex hull of $B_1(\epsilon_1) \cup B_\infty(\epsilon_\infty)$. If $d \geq 2$ and $\epsilon_1 \in (\epsilon_\infty, d\epsilon_\infty)$, then*

$$\min_{x \in \mathbb{R}^d \setminus C} \|x\|_p = \frac{\epsilon_1}{\left(\epsilon_1/\epsilon_\infty - \alpha + \alpha^q\right)^{1/q}}, \tag{3}$$

*where $\alpha = \frac{\epsilon_1}{\epsilon_\infty} - \lfloor \frac{\epsilon_1}{\epsilon_\infty} \rfloor$ and $\frac{1}{p} + \frac{1}{q} = 1$.*

As standard architectures using ReLU activation function yield a piecewise affine classifier one can anticipate that this result gives at least a rule of thumb on the expected $l_p$-robustness when one is $l_1$- and $l_\infty$-robust. Again with the choice of $\epsilon_1, \epsilon_\infty$ from above one gets for the radius of the $l_2$-ball that fits into the convex hull $C$ of the union of $B_1(\epsilon_1)$ and $B_\infty(\epsilon_\infty)$:

$$\epsilon_2 := \min_{x \in \mathbb{R}^d \setminus C} \|x\|_2 = \frac{\epsilon_1}{\sqrt{\epsilon_1/\epsilon_\infty - \alpha + \alpha^2}} \approx 0.6178. \tag{4}$$

Thus for a desired $l_2$-robustness with radius less than $0.6178$ it is sufficient for an affine classifier, and at least plausible for a ReLU network, to enforce $l_1$-robustness with $\epsilon_1 = 12$ and $l_\infty$-robustness with $\epsilon_\infty = \frac{8}{255}$. This motivates our extreme norms adversarial training (E-AT) and fine-tuning.

## 3.2 Extreme norms adversarial training (E-AT)

In light of the geometrical argument presented in the previous section, we propose to train only on adversarial perturbations for the $l_\infty$- and $l_1$-threat models if the $l_2$-radius obtained from Theorem 3.1 is larger than the radius $\epsilon_2$ of the $l_2$-threat model. In this case it is sufficient to just train for the extremes $l_1$ and $l_\infty$ in order to achieve robustness also to the intermediate $l_p$-attacks with $p \in (1, \infty)$. Since we seek a method as expensive as standard adversarial training, for each batch we do either the $l_1$- or the $l_\infty$-attack. For full training from a random initialization simply alternating or sampling

Table 1: **CIFAR-10 - Comparison of different full training schemes:** We train WideResNet-28-10 with TRADES-XENT loss (except for MNG-AC which we use as originally proposed) for each scheme (repeated for 3 seeds), and report the robust accuracy wrt $l_\infty$, $l_2$, $l_1$ and the union of the threat models. Moreover, we show the clean accuracy and the time per epoch of training.

| method | clean | $l_\infty$ ($\epsilon_\infty = \frac{8}{255}$) | $l_2$ ($\epsilon_2 = 0.5$) | $l_1$ ($\epsilon_1 = 12$) | union | time/epoch |
|---|---|---|---|---|---|---|
| $l_\infty$-AT | $82.6 \pm 0.52$ | $52.0 \pm 0.70$ | $59.7 \pm 0.22$ | $9.1 \pm 0.22$ | $9.1 \pm 0.22$ | 922 s |
| $l_2$-AT | $88.2 \pm 0.37$ | $35.9 \pm 0.17$ | $70.9 \pm 0.39$ | $36.1 \pm 0.25$ | $31.3 \pm 0.17$ | 928 s |
| $l_1$-AT | $83.7 \pm 0.16$ | $30.7 \pm 0.74$ | $65.1 \pm 0.50$ | $61.6 \pm 0.34$ | $30.7 \pm 0.74$ | 949 s |
| SAT | $80.5 \pm 0.57$ | $45.9 \pm 0.46$ | $66.7 \pm 0.29$ | $55.9 \pm 0.49$ | $45.7 \pm 0.62$ | 925 s |
| MNG-AC | $81.3 \pm 0.33$ | $43.5 \pm 0.66$ | $66.9 \pm 0.22$ | $57.6 \pm 0.84$ | $43.3 \pm 0.70$ | 1500 s |
| AVG | $82.5 \pm 0.41$ | $45.4 \pm 1.11$ | $68.0 \pm 0.87$ | $55.0 \pm 0.25$ | $45.1 \pm 1.06$ | 2771 s |
| MAX | $79.9 \pm 0.14$ | $48.4 \pm 0.74$ | $65.3 \pm 0.29$ | $50.2 \pm 0.59$ | $47.4 \pm 0.77$ | 2479 s |
| MSD | $80.6 \pm 0.33$ | $48.0 \pm 0.19$ | $65.6 \pm 0.33$ | $51.7 \pm 0.39$ | $46.9 \pm 0.09$ | 1554 s |
| E-AT unif. | $79.7 \pm 0.17$ | $45.4 \pm 0.50$ | $66.0 \pm 0.46$ | $55.6 \pm 0.54$ | $45.1 \pm 0.65$ | 939 s |
| E-AT | $79.9 \pm 0.69$ | $46.6 \pm 0.24$ | $66.2 \pm 0.61$ | $56.0 \pm 0.37$ | $46.4 \pm 0.28$ | 921 s |

uniformly at random from the $l_1$- and $l_\infty$-attack, works already well. However, for very quick fine-tuning, e.g. just one epoch in the case of ImageNet, for multiple norm robustness from an existing classifier robust wrt a single threat model, one has to take into account the existing robustness of the model. Thus we use an adaptive sampling strategy based on the running averages, reset at every epoch, of the robust training errors $\text{rerr}_1$ and $\text{rerr}_\infty$ (note that these running averages are computed just from averaging the robust error on the batches where the respective attack has been performed, thus no extra attacks are necessary), such that the probability for sampling the $l_p$-threat model is

$$\frac{\text{rerr}_p}{\text{rerr}_1 + \text{rerr}_\infty}, \quad \text{for } p \in \{1, \infty\}. \tag{5}$$

The motivation for this sampling scheme is that the robust error in the union $\Delta$ is mainly influenced by the worst threat model. We show the effect of the biased sampling in E-AT fine-tuning in Sec. C.3. The next section shows that E-AT is a very competitive baseline compared to the significantly more involved and up to three times more expensive training schemes discussed in Sec. 2.1.

### 3.3 MULTIPLE NORM ROBUSTNESS FROM RANDOM INITIALIZATION

In this experiment on CIFAR-10 we evaluate the performance of the different adversarial training variants to achieve adversarial robustness wrt the union of the $l_\infty$, $l_2$ and $l_1$-threat models where we use the standard radii $\epsilon_1 = 12$, $\epsilon_2 = 0.5$ and $\epsilon_\infty = \frac{8}{255}$. As baselines we report the $l_p$-robustness of models specifically trained for only a single norm: $l_1$-AT, $l_2$-AT and $l_\infty$-AT. We train WideResNet-28-10 (Zagoruyko & Komodakis, 2016) with the TRADES-XENT loss since Gowal et al. (2020) show that this yields slightly higher robustness than standard TRADES (Zhang et al., 2019) without additional data (see Sec. A for more details). For $l_p$-AT, SAT and E-AT we use APGD (Croce & Hein, 2020c) for training, while for AVG, MAX and MSD we use standard PGD for $l_2, l_\infty$ and SLIDE (Tramèr & Boneh, 2019) for $l_1$-PGD, even though we have observed no strong difference for multiple norms training compared to APGD. However, note that for MSD we do not use the $l_1$-PGD variant suggested by Maini et al. (2020) as this resulted in significantly worse performance (worse than all other methods). For MNG-AC (Madaan et al., 2021) we use the original code (with the $\epsilon_1$ we consider here and rescaled step size). Finally, we show also the performance of E-AT using a uniformly random sampling scheme (named E-AT unif.) for selecting the $l_p$-attack to use for each batch rather than the biased one presented in Eq. (5).

In the results in Table 1 we can see that the models we retrained using MSD and MAX perform best with $46.9\%$ and $47.4\%$ adversarial robustness in the union, but our E-AT attains very close results with $46.4\%$ (within the range of standard deviation) with $1.7\times$ and $2.7\times$ faster runtime per epoch. Given that a pure $l_\infty$-training yields $52.0\%$ $l_\infty$-robustness which turns out to be the most difficult norm, it is quite remarkable that multiple-norm robustness can be achieved with a relatively minor loss. The other variants SAT, MNG-AT, AVG and E-AT unif. perform worse with higher or similar computational cost to E-AT. Note that E-AT unif. performs similarly to SAT (which also samples uniformly the $l_p$-attack for each batch) in every threat model without seeing $l_2$-attacks at training

Table 2: **CIFAR-10 - 3 epochs of E-AT fine-tuning on $l_p$-robust models:** We use E-AT to fine-tune models robust wrt a single $l_p$-norm for multiple-norm robustness, and report the robust accuracy on 1000 test points for all threat models, and the difference compared to the initial classifier. (*) indicates that additional data is used.

| model | | clean | | $l_\infty$ ($\epsilon_\infty = \frac{8}{255}$) | | $l_2$ ($\epsilon_2 = 0.5$) | | $l_1$ ($\epsilon_1 = 12$) | | union | |
|---|---|---|---|---|---|---|---|---|---|---|---|
| **Fine-tuning $l_\infty$-robust models** | | | | | | | | | | | |
| RN-50 - $l_\infty$ | | 88.7 | | 50.9 | | 59.4 | | 5.0 | | 5.0 | |
| (Engstrom et al., 2019) + FT | | 86.2 | -2.5 | 46.0 | -4.9 | 70.1 | 10.7 | 49.2 | 44.2 | 43.4 | 38.4 |
| WRN-34-20 - $l_\infty$ | | 87.2 | | 56.6 | | 63.7 | | 8.5 | | 8.5 | |
| (Gowal et al., 2020) + FT | | 88.3 | 1.1 | 49.3 | -7.3 | 71.8 | 8.1 | 51.2 | 42.7 | 46.2 | 37.7 |
| WRN-28-10 - $l_\infty$ (*) | | 90.3 | | 59.1 | | 65.7 | | 8.0 | | 8.0 | |
| (Carmon et al., 2019) + FT | | 90.3 | 0.0 | 52.6 | -6.5 | 74.7 | 9.0 | 54.0 | 46.0 | 48.7 | 40.7 |
| WRN-28-10 - $l_\infty$ (*) | | 89.9 | | 62.9 | | 67.2 | | 10.8 | | 10.8 | |
| (Gowal et al., 2020) + FT | | 91.2 | 1.3 | 53.9 | -9.0 | 76.0 | 8.8 | 56.9 | 46.1 | 50.1 | 39.3 |
| WRN-70-16 - $l_\infty$ (*) | | 90.7 | | 65.6 | | 66.9 | | 8.1 | | 8.1 | |
| (Gowal et al., 2020) + FT | | 91.6 | 0.9 | 54.3 | -11.3 | 78.2 | 11.3 | 58.3 | 50.2 | 51.2 | 43.1 |
| **Fine-tuning $l_2$-robust models** | | | | | | | | | | | |
| RN-50 - $l_2$ | | 91.5 | | 29.7 | | 70.3 | | 27.0 | | 23.0 | |
| (Engstrom et al., 2019) + FT | | 87.8 | -3.7 | 43.1 | 13.4 | 70.8 | 0.5 | 50.2 | 23.2 | 41.7 | 18.7 |
| RN-50 - $l_2$ (*) | | 91.1 | | 37.7 | | 73.4 | | 31.2 | | 28.8 | |
| (Augustin et al., 2020) + FT | | 87.0 | -4.1 | 47.2 | 9.5 | 70.4 | -3.0 | 54.1 | 22.9 | 46.0 | 17.2 |
| WRN-70-16 - $l_2$ (*) | | 94.1 | | 43.1 | | 81.7 | | 34.6 | | 32.4 | |
| (Gowal et al., 2020) + FT | | 91.2 | -2.9 | 51.9 | 8.8 | 79.2 | -2.5 | 58.8 | 24.2 | 49.7 | 17.3 |
| **Fine-tuning $l_1$-robust models** | | | | | | | | | | | |
| RN-18 - $l_1$ | | 87.1 | | 22.0 | | 64.8 | | 60.3 | | 22.0 | |
| (Croce & Hein, 2021) + FT | | 83.5 | -3.6 | 40.3 | 18.3 | 68.1 | 3.3 | 55.7 | -4.6 | 40.1 | 18.1 |

time. Thus E-AT provides a very efficient and effective alternative to MSD and MAX which allows to scale adversarial training for multiple norms to larger problems such as ImageNet. We repeat the comparison with the same setup but on the smaller PreAct ResNet-18 (He et al., 2016) in Sec. B.3 with standard adversarial training on the cross-entropy loss and obtain similar results. Thus, E-AT generalizes across architectures and training schemes. Finally, we observe how almost all methods for multiple norms lead to a drop of clean accuracy compared to the single norm $l_p$-AT, showing that the threat model of the union is significantly more challenging.

### 3.4 Multiple-norm robustness via fast fine-tuning of existing robust models

In the previous section, we have shown that training for only the two extreme threat models is sufficient to achieve robustness wrt all three norms. Prior works (Tramèr & Boneh, 2019; Kang et al., 2019a) observed that models adversarially trained wrt $l_\infty$ give non trivial robustness to $l_2$-attacks, although lower than what one gets directly training against such attacks, and vice versa. This is confirmed by our evaluation of the models in Table 1 (top part), where we also notice that $l_1$-AT provides good robust accuracy in $l_2$. On the other hand, training for $l_\infty$ resp. $l_1$ does not yield particular robustness to the dual norm, which is reasonable since the perturbations generated in the two threat models are very different, while $l_2$ can be seen as an intermediate case which can defend at least partially against $l_\infty$- and $l_1$-attacks. Therefore, we propose to use models trained for robustness wrt a single norm as good initializations to achieve, within a small computational budget, multiple norms robustness. This is done by fine-tuning pretrained models with our E-AT for 3 epochs when using CIFAR-10 and 1 epoch with ImageNet-1k, starting with learning rate 0.05 or 0.01, depending on the model, and decreasing by a factor of 10 at each epoch. We do 10 steps of APGD in adversarial training for CIFAR-10, while 5 and 15 with $l_\infty$ and $l_1$ respectively on ImageNet since optimizing in

Table 3: **ImageNet - Results of one epoch of E-AT fine-tuning of existing robust models:** We use existing models trained to be robust wrt a single $l_p$-ball (either $l_\infty$ or $l_2$) from Engstrom et al. (2019) and fine-tune them for a single epoch with our E-AT scheme.

| model | clean | | $l_\infty$ ($\epsilon_\infty = \frac{4}{255}$) | | $l_2$ ($\epsilon_2 = 2$) | | $l_1$ ($\epsilon_1 = 255$) | | union | |
|---|---|---|---|---|---|---|---|---|---|---|
| RN-50 - $l_\infty$ | 62.9 | | 29.8 | | 17.7 | | 0.0 | | 0.0 | |
| (Engstrom et al., 2019) + FT | 58.0 | -4.9 | 27.3 | -2.5 | 41.1 | 23.4 | 24.0 | 24.0 | 21.7 | 21.7 |
| RN-50 - $l_2$ | 58.7 | | 25.0 | | 40.5 | | 14.0 | | 13.5 | |
| (Engstrom et al., 2019) + FT | 56.7 | -2.0 | 26.7 | 1.7 | 41.0 | 0.5 | 25.4 | 11.4 | 23.1 | 9.6 |

the $l_1$-ball requires more iterations in that case. When the model was originally trained with extra data beyond the training set on CIFAR-10, we use the 500k images introduced by Carmon et al. (2019) as additional data for fine-tuning (see also Sec. A).

**CIFAR-10:** RobustBench (Croce et al., 2020a) provides a collection of the currently most robust classifiers. We took the most robust models, among those which do not use synthetic data, for $l_2$- and $l_\infty$-norm and the $l_1$-robust one from Croce & Hein (2021) (all are trained with the same radii $\epsilon_p$ as in our experiment). Each of them is fine-tuned for 3 epochs with E-AT where ($*$) denotes models trained using extra data, which also means that we used the extra data from Carmon et al. (2019) for fine-tuning (Gowal et al. (2020) have created their own extra data but which is not available). We present in Table 2 the results. First of all the fine-tuning works for all tested architectures and results in many cases in stronger robustness in the union than for the specifically trained WideResNet-28-10 models. In particular, the most robust $l_\infty$-model from Gowal et al. (2020) with $65.6\%$ $l_\infty$-robustness and only $8.1\%$ $l_1$-robustness can be fine-tuned to a multiple-norm robust model with $51.2\%$ robustness which is up to our knowledge the best reported multiple-norm robustness. Very interesting is that the $l_2$-robustness of $78.2\%$ is quite close to the $81.7\%$ $l_2$-robustness of the specifically $l_2$-trained model from Gowal et al. (2020). Moreover, the $l_1$-robustness of $58.3\%$ is close to the best reported one of $60.3\%$ from Croce & Hein (2021) (however we improve this a lot in the next section) and the model has even higher clean accuracy. Clearly, this comes at the price of a significant loss in $l_\infty$ but this is to be expected. Striking is that fine-tuning the $l_2$-robust model from Gowal et al. (2020) results in a very similar result. In a nutshell, E-AT fine-tuning of existing $l_p$-robust models yields very efficient and competitive baselines for future research in this area.

**ImageNet:** Similarly to CIFAR-10 we start with the specific $l_2$-resp. $l_\infty$-robust models from Engstrom et al. (2019) which have been trained with $\epsilon_2 = 3$ and $\epsilon_\infty = \frac{4}{255}$. We use $\epsilon = 2$ for the experiments as the robust accuracy is still in a reasonable range of $40\%$ and together with our choice of $\epsilon_1 = 255$ and $\epsilon_\infty = \frac{4}{255}$ the $l_2$-radius from Theorem 3.1 is almost exactly 2. We fine-tune the two models only for a single epoch and report the results in Table 3. Again, we note that our evaluation is stronger than the original one (although this is on a different number of data points) which effects in particular the $l_\infty$-robustness. Note that the initial $l_\infty$-model is completely non-robust for $l_1$ but achieves $24.0\%$ $l_1$-robust accuracy and also the $l_2$-robust accuracy improves from $17.1\%$ to $41.1\%$ and this at the price of a relatively small loss in $l_\infty$-robust and clean accuracy. For the $l_2$-robust model all robust accuracies improve as the original model was trained for $\epsilon = 3$. It yields the best multiple-norm robust accuracy of $23.1\%$. Up to our knowledge no multiple-norm robustness has been reported before for ImageNet and thus these results are an important baseline.

**Additional experiments:** Sec. C in the appendix contains further studies about E-AT fine-tuning: we show that fine-tuning a naturally trained model does not provide competitive robustness and leads to low clean accuracy. Moreover, we show the stability of the scheme over random seeds, and that increasing the number of epochs progressively improves the robustness in the union.

## 4  FINE-TUNING $l_p$-ROBUST MODELS TO BECOME $l_q$-ROBUST FOR $p \neq q$

Motivated by our results for the multiple-norm threat model we study to which extent we can fine-tune an $l_p$-robust model to a $l_q$-robust model with $p \neq q$. Again the emphasis is on an extremely short fine-tuning time so that this is much faster than full adversarial training.

Table 4: **Fine-tuning $l_p$-robust models to another threat model:** For each norm we fine-tune the most robust models wrt the other ones for 3 epochs for CIFAR-10 and 1 epoch for ImageNet and report clean and robust accuracy for all threat models. Even for the threat models where the robustness of the original model is low, the fine-tuning is sufficient to yield robustness almost at the same level of the specialized models with same architecture. For each threat model (column) we highlight in blue the model trained for the specific norm, in orange those only fine-tuned in the target norm. The values of the thresholds $\epsilon$ are the same used the multiple norms experiments.

| **CIFAR-10** | clean | $l_\infty$ | $l_2$ | $l_1$ | **ImageNet** | clean | $l_\infty$ | $l_2$ | $l_1$ |
|---|---|---|---|---|---|---|---|---|---|
| WRN-70-16 (Gowal et al., 2020) - $l_\infty$ (*) | | | | | RN-50 (Engstrom et al., 2019) - $l_\infty$ | | | | |
| original | 90.7 | 65.6 | 66.9 | 8.1 | original | 62.9 | 29.8 | 17.7 | 0.0 |
| + FT wrt $l_2$ | 92.8 | 47.4 | 80.0 | 34.0 | + FT wrt $l_2$ | 62.9 | 25.5 | 41.5 | 8.4 |
| + FT wrt $l_1$ | 92.4 | 33.9 | 74.7 | **70.2** | + FT wrt $l_1$ | 57.7 | 18.0 | 37.6 | 27.4 |
| WRN-70-16 (Gowal et al., 2020) - $l_2$ (*) | | | | | RN-50 (Engstrom et al., 2019) - $l_2$ | | | | |
| original | 94.1 | 43.1 | 81.7 | 34.6 | original | 58.7 | 25.0 | 40.5 | 14.0 |
| + FT wrt $l_\infty$ | 92.3 | 58.5 | 73.5 | 11.4 | + FT wrt $l_\infty$ | 59.1 | 31.5 | 40.1 | 7.5 |
| + FT wrt $l_1$ | 92.8 | 29.2 | 75.7 | 68.9 | + FT wrt $l_1$ | 56.8 | 18.0 | 37.1 | **28.7** |
| RN-18 (Croce & Hein, 2021) - $l_1$ | | | | | | | | | |
| original | 87.1 | 22.0 | 64.8 | 60.3 | | | | | |
| + FT wrt $l_\infty$ | 82.7 | 44.2 | 66.6 | 25.4 | | | | | |
| + FT wrt $l_2$ | 88.0 | 31.0 | 69.8 | 39.7 | | | | | |

**CIFAR-10:** We fine-tune for 3 epochs the most $l_\infty$-robust model at $\epsilon_\infty = \frac{8}{255}$ of Gowal et al. (2020) with adversarial training wrt $l_2$ and $l_1$ with $\epsilon_2 = 0.5$ and $\epsilon_1 = 12$. Table 4 shows that fine-tuning for $l_1$-robustness yields 70.2% $l_1$-robust accuracy which is 9.9% more than the previously most robust model. Also we get a strikingly high $l_2$-robust accuracy for the $l_2$-fine-tuned model of 80.0% not far away from the 81.7% which ones gets by training for $l_2$ from scratch. Surprisingly, fine-tuning the $l_2$-robust model of Gowal et al. (2020) wrt $l_1$ does not outperform the $l_1$-robustness achieved by fine-tuning their $l_\infty$-robust model. Interestingly, fine-tuning the $l_1$-robust PreAct ResNet-18 for $l_2$ yields a better $l_2$-robustness than $l_2$-training from scratch, see Table 5. Again this shows that fine-tuning of existing models is minimal effort and already provides strong baselines for adversarial robustness obtained by adversarial training from scratch and in some cases even outperforms them.

**ImageNet:** We fine-tune the $l_2$- and $l_\infty$-robust models from Engstrom et al. (2019) to the other threat model respectively and wrt $l_1$. The results are in Table 4. The resulting classifiers are both more robust than the originally trained models from Engstrom et al. (2019): we get a $l_2$-robust accuracy of 41.5% which is 0.6% higher than the $l_2$-model and a $l_\infty$-robust accuracy of 31.5% which is 1.7% better than the original $l_\infty$-model. For both models it is possible to achieve within one epoch of fine-tuning a non-trivial robustness in $l_1$. In particular, notice that the model for $l_\infty$ initially has 0.0% robustness wrt $l_1$, while after fine-tuning it achieves 27.4% robust accuracy which is similar to what obtained by fine-tuning the $l_2$-robust model for $l_1$ which yields 28.7%. Note that when we fine-tuned for multiple-norm robustness we got a $l_1$-robust accuracy of 21.7% resp. 23.1% (Table 3). Thus by fine-tuning specifically for $l_1$-robustness one gains 5.7% resp. 5.6% $l_1$-robust accuracy. Up to our knowledge our ImageNet models are the first ones for which $l_1$-robustness is reported.

## 5 CONCLUSION

Based on the geometry of the $l_p$-balls we have introduced E-AT, a novel training scheme for multiple norm robustness which achieves comparable adversarial robustness in the union while being significantly faster. We also show for the first time that fine-tuning can be used to transfer adversarial robustness from a single $l_p$-threat model to the multiple norms one, and that one can even obtain an $l_q$-robust classifier with a quick fine-tuning of an $l_p$-robust one with $p \neq q$. This yields strong baselines for future research. We have in this way generated models with SOTA performance for multiple-norm and $l_1$-robustness on CIFAR-10 and the first models on ImageNet which show significant multiple-norm as well as $l_1$-robustness.

ETHICS STATEMENT

There are no conflicts of interest in this work. As we consider the efficient training and fine-tuning of adversarially robust models, we think that this work contributes to more trustworthy AI systems and thus has rather positive implications.

REPRODUCIBILITY STATEMENT

We provide information about training in Sec. 2.1 and Sec. 4 of the main part, and add further details in Sec. A of the appendix. We provide in the supplementary material an implementation of our method in PyTorch. Note that most of the models used for fine-tuning are available in RobustBench, and the evaluation code is part of AutoAttack. We report for most of the experiments the mean and standard deviations over different random seeds, as well as the runtime and infrastructure used.

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

## A    EXPERIMENTAL DETAILS

For the comparison of training schemes we use for multiple-norm robustness we train PreAct ResNet-18 (He et al., 2016) with softplus activation function for 80 epochs with initial learning rate of 0.05 reduced by a factor of 10 after 70 epochs. When training WideResNet-28-10 (Zagoruyko & Komodakis, 2016) we use a cyclic schedule for the learning rate with maximum value 0.1 for 30 epochs. We use SGD optimizer with momentum of 0.9 and weight decay of $5 \cdot 10^{-4}$, batch size of 128. We use random cropping and horizontal flipping as augmentation techniques. For adversarial training of models robust wrt a single norm and with SAT and our novel scheme E-AT we use APGD with default parameters, while for the retrained AVG, MAX, and MSD we use PGD for $l_\infty$ (step size $\epsilon_\infty/4$) and $l_2$ (step size $\epsilon/3$), SLIDE (Tramèr & Boneh, 2019) for $l_1$ (standard parameters). For all methods we use 10 steps for the inner maximization problem in adversarial training (note that AVG and MAX repeat the attack for all threat models, and MSD tests multiple steps, thus they are more expensive). For all schemes we select the best performing checkpoint for the comparison when using the piecewise schedule, the final checkpoint with the cyclic schedule. Moreover, we use the TRADES-XENT loss (TRADES loss (Zhang et al., 2019) with adversarial points maximizing the cross-entropy loss) since Gowal et al. (2020) show that this gives slightly better robustness on CIFAR-10 without additional data, while we use standard adversarial training (Madry et al., 2018) for PreAct ResNet-18. We train the classifiers of MNG-AC (Madaan et al., 2021) with the original code, where we set $\epsilon_1 = 12$ and rescale the step size linearly. Finally, for the runtime comparison we run each method on a single Tesla V100 GPU.

For fine-tuning on CIFAR-10 we use 3 epochs and the same setup as for full training except for the learning rate schedule, since in this case we use as initial value the best performing one in $\{0.01, 0.05\}$ (the larger value works best for the smaller networks) and reduce it by a factor of 10 at the beginning of each epoch. When the model was originally trained with extra data beyond the training set on CIFAR-10, we use the 500k images introduced by Carmon et al. (2019) as additional data for fine-tuning, and each batch is splitted equally between standard and extra images, and we count 1 epoch when the whole standard training set has been used: note that in this way, using only 3 epochs not the whole pseudo-labelled dataset is exploited.

For fine-tuning on ImageNet we use 1 epoch, initial learning rate of 0.01, reduced by a factor of 10 every $1/3$ of training steps. We follow the setup of Engstrom et al. (2019) for data augmentation and setting batch size to 256 and weight decay to $10^{-4}$. For adversarial training we use APGD with 5 steps for $l_\infty$ and $l_2$, 15 steps for $l_1$ since optimizing in the $l_1$-ball intersected with the box constraints is more challenging, see Croce & Hein (2021).

## B    ADDITIONAL ANALYSIS, EVALUATION AND EXPERIMENTS FOR E-AT

We here analyze in more details our E-AT scheme and expand the comparison to existing methods presented above.

### B.1    ROBUSTNESS WRT $l_2$ OF E-AT

To show the effect of E-AT on $l_2$-robustness we plot in Fig. 3 the robust accuracy wrt $l_2$ computed with FAB (Croce & Hein, 2020a), which minimizes the size of the perturbations, as a function of the threshold $\epsilon_2$ for a PreAct ResNet-18 trained with $l_2$-AT at $\epsilon_2 = 0.5$ and one using E-AT (see complete results for such models in Sec. B.3). Theorem 3.1 suggests that the extreme norms training provides robustness at $\epsilon_2 \approx 0.62$, which is confirmed by the plots. Although no $l_2$-attack has been used during training by the E-AT model, it has robustness wrt $l_2$ similar to that of the classifier specifically trained for such threat model.

### B.2    CONFIRMATION THAT ADVERSARIAL TRAINING WITH THE EXTREME NORMS IS SUFFICIENT: ANALYSIS OF MAX AND MSD TRAINING

Both MAX and MSD schemes perform adversarial training considering all the threat models simultaneously, and we analyze here their training procedure in more detail. Fig. 4 shows for the MAX strategy how many times, in percentage, for each epoch the points computed for each threat model realize the maximal loss over the three attacks $(l_1, l_2, l_\infty)$, and then are subsequently used for the

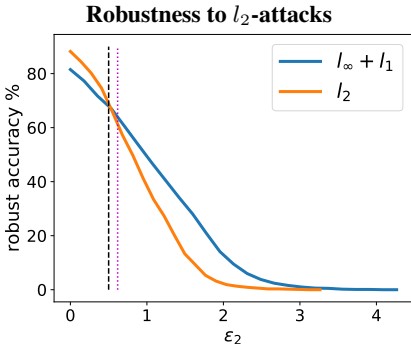

Figure 3: $l_2$-robustness curve of a model trained with $l_2$-adversarial training (AT) for $\epsilon_2 = 0.5$ (orange) versus our E-AT (blue), which is expected to yield robustness at $\epsilon_2 = 0.62$. **Although $l_2$-attacks are not used for training, our extreme-adv. training scheme E-AT yields $l_2$-robustness similar to the one obtained with specific $l_2$-adversarial training.**

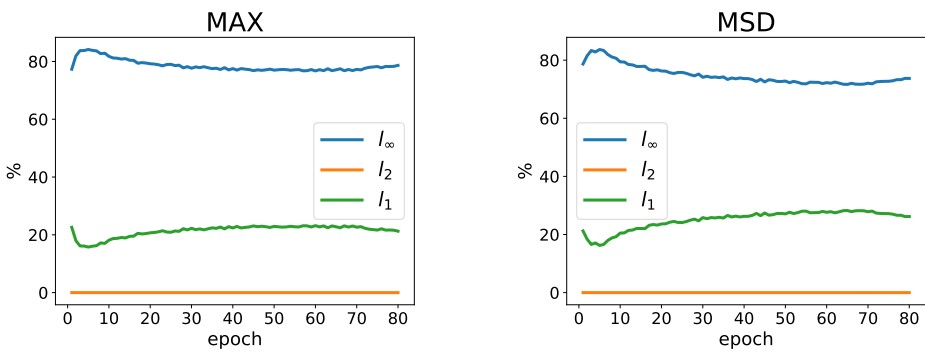

Figure 4: **CIFAR-10, ResNet18. Left:** For MAX-training we show for each epoch during training the percentage of points attaining for the indicated $l_p$-threat model ($p \in \{1, 2, \infty\}$) the highest loss over the three threat models. **Right:** For MSD-training we show the percentage of steps taken wrt each threat model over epochs (note that MSD does the steepest descent step for each $l_p$-threat model and then realizes the one yielding maximal loss).

update of the model. Similarly, for MSD, we show the frequency with which a step wrt each $l_p$-norm is taken when computing the adversarial points (average over all iterations and training points). In both cases the $l_\infty$-threat model is the most used one with the $l_1$-threat model being used $3 - 4$ times less often. However, the $l_2$-threat model is almost never chosen. This empirically confirms the analysis from Theorem 3.1 which shows that training only wrt $l_\infty$ and $l_1$ is (at least for a linear classifier) sufficient to achieve $l_2$-robustness for the chosen $\epsilon_2$ and thus during training no extra updates wrt the $l_2$-threat model are necessary. This is in line with the results reported in Fig. 3 which show that for different thresholds the $l_2$-robustness achieved by E-AT-training is similar to that of standard $l_2$-training (for generating Fig. 3 we use FAB attack (Croce & Hein, 2020a) to compute robust accuracy at varying $\epsilon_2$).

## B.3 E-AT FROM RANDOM INITIALIZATION WITH DIFFERENT ARCHITECTURE

We repeat the experiment from the main part about training for multiple norms robustness from random initialization with a smaller architecture like PreAct ResNet-18 (He et al., 2016). MAX-training yields the most robust model but E-AT is only $1.6\%$ worse outperforming SAT and AVG but note that the standard deviation is rather high. Since it has the same architecture we additionally include the original MSD model of Maini et al. (2020), marked with (*), which obtains $41.4\%$ robustness in the union, whereas with our reimplementation we get $43.9\%$, improving their results significantly. Note that they reported in their paper $47.0\%$ robustness in the union, while our APGD-

Table 5: **CIFAR-10 - Comparison of different full training schemes for multiple-norm robustness on PreAct ResNet-18:** For each scheme we report the robust accuracy wrt $l_\infty$, $l_2$, $l_1$ and the worst case over the union of the threat models. Moreover, we show the clean accuracy and the time per epoch of training. MSD and MAX perform best in the union, but our E-AT achieves almost the same robustness in the union but is better in $l_1$- and $l_2$-robustness and requires only about about a third or half of the training time. (*) indicates the original MSD model from Maini et al. (2020).

| method | clean | $l_\infty$ ($\epsilon_\infty = \frac{8}{255}$) | $l_2$ ($\epsilon_2 = 0.5$) | $l_1$ ($\epsilon_1 = 12$) | union | time/epoch |
|---|---|---|---|---|---|---|
| $l_\infty$-AT | $84.0 \pm 0.31$ | $48.1 \pm 0.21$ | $59.7 \pm 0.41$ | $6.3 \pm 0.97$ | $6.3 \pm 0.97$ | 151 s |
| $l_2$-AT | $88.9 \pm 0.57$ | $27.3 \pm 1.79$ | $68.7 \pm 0.09$ | $25.3 \pm 1.60$ | $20.9 \pm 1.80$ | 153 s |
| $l_1$-AT | $85.9 \pm 1.07$ | $22.1 \pm 0.14$ | $64.9 \pm 0.49$ | $59.5 \pm 0.85$ | $22.1 \pm 0.09$ | 195 s |
| SAT | $83.9 \pm 0.82$ | $40.7 \pm 0.71$ | $68.0 \pm 0.39$ | $54.0 \pm 1.20$ | $40.4 \pm 0.66$ | 161 s |
| AVG | $84.6 \pm 0.31$ | $40.8 \pm 0.66$ | $68.4 \pm 0.71$ | $52.1 \pm 0.37$ | $40.1 \pm 0.78$ | 479 s |
| MAX | $80.4 \pm 0.54$ | $45.7 \pm 0.90$ | $66.0 \pm 0.41$ | $48.6 \pm 0.82$ | $44.0 \pm 0.71$ | 466 s |
| MSD (*) | 82.1 | 43.1 | 64.5 | 46.5 | 41.4 | - |
| MSD | $81.1 \pm 1.14$ | $44.9 \pm 0.63$ | $65.9 \pm 0.64$ | $49.5 \pm 1.18$ | $43.9 \pm 0.76$ | 306 s |
| E-AT unif. | $82.2 \pm 1.84$ | $42.7 \pm 0.74$ | $67.5 \pm 0.46$ | $53.6 \pm 0.12$ | $42.4 \pm 0.60$ | 163 s |
| E-AT | $81.9 \pm 1.44$ | $43.0 \pm 0.87$ | $66.4 \pm 0.58$ | $53.0 \pm 0.29$ | $42.4 \pm 0.73$ | 160 s |

based evaluation reduces this to 41.4% which shows that our robustness evaluation is significantly stronger.

## B.4 ADDITIONAL STATISTICS

Table 6: **CIFAR-10 - Comparison of different full training schemes:** We repeat the results from Table 1 and Table 5 with additionally the average robust accuracy over the three threat models (last column).

| method | clean | $l_\infty$ ($\epsilon_\infty = \frac{8}{255}$) | $l_2$ ($\epsilon_2 = 0.5$) | $l_1$ ($\epsilon_1 = 12$) | union | average |
|---|---|---|---|---|---|---|
| **WideResNet-28-10** | | | | | | |
| $l_\infty$-AT | $82.6 \pm 0.52$ | $52.0 \pm 0.70$ | $59.7 \pm 0.22$ | $9.1 \pm 0.22$ | $9.1 \pm 0.22$ | $40.3 \pm 0.4$ |
| $l_2$-AT | $88.2 \pm 0.37$ | $35.9 \pm 0.17$ | $70.9 \pm 0.39$ | $36.1 \pm 0.25$ | $31.3 \pm 0.17$ | $47.6 \pm 0.2$ |
| $l_1$-AT | $83.7 \pm 0.16$ | $30.7 \pm 0.74$ | $65.1 \pm 0.50$ | $61.6 \pm 0.34$ | $30.7 \pm 0.74$ | $52.5 \pm 0.5$ |
| SAT | $80.5 \pm 0.57$ | $45.9 \pm 0.46$ | $66.7 \pm 0.29$ | $55.9 \pm 0.49$ | $45.7 \pm 0.62$ | $56.2 \pm 0.4$ |
| MNG-AC | $81.3 \pm 0.33$ | $43.5 \pm 0.66$ | $66.9 \pm 0.22$ | $57.6 \pm 0.84$ | $43.3 \pm 0.70$ | $56.0 \pm 0.4$ |
| AVG | $82.5 \pm 0.41$ | $45.4 \pm 1.11$ | $68.0 \pm 0.87$ | $55.0 \pm 0.25$ | $45.1 \pm 1.06$ | $56.1 \pm 0.7$ |
| MAX | $79.9 \pm 0.14$ | $48.4 \pm 0.74$ | $65.3 \pm 0.29$ | $50.2 \pm 0.59$ | $47.4 \pm 0.77$ | $54.6 \pm 0.5$ |
| MSD | $80.6 \pm 0.33$ | $48.0 \pm 0.19$ | $65.6 \pm 0.33$ | $51.7 \pm 0.39$ | $46.9 \pm 0.09$ | $55.1 \pm 0.2$ |
| E-AT unif. | $79.7 \pm 0.17$ | $45.4 \pm 0.50$ | $66.0 \pm 0.46$ | $55.6 \pm 0.54$ | $45.1 \pm 0.65$ | $55.7 \pm 0.4$ |
| E-AT | $79.9 \pm 0.69$ | $46.6 \pm 0.24$ | $66.2 \pm 0.61$ | $56.0 \pm 0.37$ | $46.4 \pm 0.28$ | $56.3 \pm 0.3$ |
| **PreAct ResNet-18** | | | | | | |
| $l_\infty$-AT | $84.0 \pm 0.31$ | $48.1 \pm 0.21$ | $59.7 \pm 0.41$ | $6.3 \pm 0.97$ | $6.3 \pm 0.97$ | $38.0 \pm 0.3$ |
| $l_2$-AT | $88.9 \pm 0.57$ | $27.3 \pm 1.79$ | $68.7 \pm 0.09$ | $25.3 \pm 1.60$ | $20.9 \pm 1.80$ | $40.5 \pm 1.1$ |
| $l_1$-AT | $85.9 \pm 1.07$ | $22.1 \pm 0.14$ | $64.9 \pm 0.49$ | $59.5 \pm 0.85$ | $22.1 \pm 0.09$ | $48.8 \pm 0.4$ |
| SAT | $83.9 \pm 0.82$ | $40.7 \pm 0.71$ | $68.0 \pm 0.39$ | $54.0 \pm 1.20$ | $40.4 \pm 0.66$ | $54.2 \pm 0.8$ |
| AVG | $84.6 \pm 0.31$ | $40.8 \pm 0.66$ | $68.4 \pm 0.71$ | $52.1 \pm 0.37$ | $40.1 \pm 0.78$ | $53.8 \pm 0.1$ |
| MAX | $80.4 \pm 0.54$ | $45.7 \pm 0.90$ | $66.0 \pm 0.41$ | $48.6 \pm 0.82$ | $44.0 \pm 0.71$ | $53.4 \pm 0.5$ |
| MSD | $81.1 \pm 1.14$ | $44.9 \pm 0.63$ | $65.9 \pm 0.64$ | $49.5 \pm 1.18$ | $43.9 \pm 0.76$ | $53.4 \pm 0.4$ |
| E-AT unif. | $82.2 \pm 1.84$ | $42.7 \pm 0.74$ | $67.5 \pm 0.46$ | $53.6 \pm 0.12$ | $42.4 \pm 0.60$ | $54.6 \pm 0.2$ |
| E-AT | $81.9 \pm 1.44$ | $43.0 \pm 0.87$ | $66.4 \pm 0.58$ | $53.0 \pm 0.29$ | $42.4 \pm 0.73$ | $54.2 \pm 0.4$ |

Table 6 repeats the results from Table 1 and Table 5 with additionally the average robust accuracy over the three threat models (last column), with standard deviation over 3 random seeds. On WRN-

28-10, in this metric E-AT achieves the best results, and in particular significantly outperforms MAX while having the same clean accuracy. For the smaller RN-18, E-AT unif. attains the highest average robustness, with E-AT and SAT being slightly worse.

## C  FINE-TUNING WITH E-AT FOR MULTIPLE NORMS ROBUSTNESS

### C.1  FINE-TUNING ROBUST AND NATURAL MODELS

Table 7: **CIFAR-10 - 3 epochs of fine-tuning with E-AT:** We report the results of fine-tuning PreAct ResNet-18 models to become robust wrt the union of the threat models. Fine-tuning any $l_p$-robust model leads to competitive clean and robust accuracy to full training, differently from using a naturally trained model.

| model | | clean | | $l_\infty$ ($\epsilon_\infty = \frac{8}{255}$) | | $l_2$ ($\epsilon_2 = 0.5$) | | $l_1$ ($\epsilon_1 = 12$) | | union | |
|---|---|---|---|---|---|---|---|---|---|---|---|
| RN-18 - standard | | 94.4 | | 0.0 | | 0.0 | | 0.0 | | 0.0 | |
| | + FT | 66.6 | -27.8 | 29.9 | 29.9 | 50.1 | 50.1 | 38.5 | 38.5 | 29.8 | 29.8 |
| RN-18 - $l_\infty$ | | 83.7 | | 48.1 | | 59.8 | | 7.7 | | 7.7 | |
| | + FT | 82.3 | -1.4 | 43.4 | -4.7 | 68.0 | 8.2 | 48.0 | 40.3 | 41.2 | 33.5 |
| RN-18 - $l_2$ | | 88.2 | | 29.8 | | 68.6 | | 27.5 | | 23.1 | |
| | + FT | 85.4 | -2.8 | 40.6 | 10.8 | 69.8 | 1.2 | 48.7 | 21.2 | 39.1 | 16.0 |
| RN-18 - $l_1$ | | 87.1 | | 22.0 | | 64.8 | | 60.3 | | 22.0 | |
| | + FT | 83.5 | -3.6 | 40.3 | 18.3 | 68.1 | 3.3 | 55.7 | -4.6 | 40.1 | 18.1 |

We fine-tune for 3 epochs PreAct ResNet-18 either naturally trained or robust wrt a single $l_p$-norm. Table 7 shows clean and robust accuracy for each threat model for the initial classifier and after E-AT fine-tuning: while for all robust models the fine-tuning yields values competitive with the full training for multiple norms (see Table 5), starting from a standard model leads to significantly lower both clean performance and robustness in the union of the three $l_p$-balls.

### C.2  RUNTIME WITH LARGE MODELS

We reported in Table 1 and Table 5 the runtime per epoch of E-AT. For larger architectures the computational cost increases significantly, and adversarial training with the WideResNet-70-16, the largest one we consider, on CIFAR-10 takes, in our experiments, around 6100 s per epoch when using only the training set and over 10000 s if the unlabelled data is used (since twice more training steps are effectively used). This shows how transfering robustness with fine-tuning might allow to obtain classifiers robust wrt different threat models fast and at much lower computational cost.

### C.3  RESULTS OVER RANDOM SEEDS AND EFFECT OF BIASED SAMPLING SCHEME

Table 8: **CIFAR-10 - Uniform vs biased sampling in E-AT for fine-tuning:** We fine-tune with E-AT for 3 epochs the RN-18 robust wrt individual norms with either uniform (E-AT unif.) or biased (E-AT) sampling scheme (mean and standard deviation of the clean and robust accuracy over 5 seeds is reported). The biased sampling scheme is helpful when fine-tuning the $l_2$ and $l_1$ models which are not robust in the most challenging threat model, i.e. $l_\infty$.

| model | clean | $l_\infty$ ($\epsilon_\infty = \frac{8}{255}$) | $l_2$ ($\epsilon_2 = 0.5$) | $l_1$ ($\epsilon_1 = 12$) | union |
|---|---|---|---|---|---|
| RN-18 - $l_\infty$ - uniform | $82.6 \pm 0.62$ | $44.4 \pm 0.37$ | $68.0 \pm 0.26$ | $48.5 \pm 0.96$ | $42.2 \pm 0.34$ |
| RN-18 - $l_\infty$ - biased | $82.7 \pm 0.41$ | $44.3 \pm 0.63$ | $68.1 \pm 0.48$ | $48.7 \pm 0.46$ | $42.2 \pm 0.78$ |
| RN-18 - $l_2$ - uniform | $85.9 \pm 0.51$ | $40.0 \pm 0.65$ | $69.4 \pm 0.68$ | $50.3 \pm 0.44$ | $38.9 \pm 0.79$ |
| RN-18 - $l_2$ - biased | $85.8 \pm 0.68$ | $40.7 \pm 0.90$ | $69.5 \pm 0.50$ | $49.5 \pm 0.54$ | $39.4 \pm 0.72$ |
| RN-18 - $l_1$ - uniform | $83.5 \pm 0.68$ | $39.8 \pm 0.51$ | $68.0 \pm 0.19$ | $55.8 \pm 0.55$ | $39.6 \pm 0.47$ |
| RN-18 - $l_1$ - biased | $83.6 \pm 0.56$ | $40.5 \pm 0.38$ | $68.1 \pm 0.13$ | $55.3 \pm 0.38$ | $40.3 \pm 0.33$ |

We study the effect of randomness in the training process when fine-tuning for robustness wrt multiple norms using E-AT. Moreover, we compare the biased sampling scheme introduced in Eq. (5) to a uniform one when to choose which $l_p$, with $p \in \{1, \infty\}$, attack to use for adversarial training for each batch. In Table 7 we report the average results and corresponding standard deviation over 5 runs with different seeds on CIFAR-10 models originally robust wrt a single norm. For all metrics and starting models different runs show similar performance (the largest standard deviation is $0.96\%$) showing the stability of the scheme. Note that in the previous experiments we used the same seed for all runs and methods, without selecting the best one in a pool. Moreover, the biased sampling schemes yields slightly better results when fine-tuning the $l_2$- and in particular the $l_1$-robust model: we hypothesize that, since $l_\infty$ is the most challenging threat model, it is important to use it more often at training time when the initial model is non robust wrt $l_\infty$.

## C.4 RESULTS USING MORE EPOCHS

Table 9: **CIFAR-10 - Fine-tuning for more epochs:** We show the effect of fine-tuning for different number of epochs (3 is the standard we use) the PreAct ResNet-18 (standard or robust wrt $l_\infty$).

| model | clean | | $l_\infty$ ($\epsilon_\infty = \frac{8}{255}$) | | $l_2$ ($\epsilon_2 = 0.5$) | | $l_1$ ($\epsilon_1 = 12$) | | union | |
|---|---|---|---|---|---|---|---|---|---|---|
| RN-18 - $l_\infty$ | 83.7 | | 48.1 | | 59.8 | | 7.7 | | 7.7 | |
| + 3 epochs FT | 82.3 | -1.4 | 43.4 | -4.7 | 68.0 | 8.2 | 48.0 | 40.3 | 41.2 | 33.5 |
| + 5 epochs FT | 83.0 | -0.7 | 45.2 | -2.9 | 68.8 | 9.0 | 50.1 | 42.4 | 43.1 | 35.4 |
| + 7 epochs FT | 83.1 | -0.6 | 44.6 | -3.5 | 68.7 | 8.9 | 50.4 | 42.7 | 42.6 | 34.9 |
| + 10 epochs FT | 84.0 | 0.3 | 44.9 | -3.2 | 69.2 | 9.4 | 51.0 | 43.3 | 42.8 | 35.1 |
| + 15 epochs FT | 84.6 | 0.9 | 44.9 | -3.2 | 69.5 | 9.7 | 52.1 | 44.4 | 43.2 | 35.5 |
| RN-18 - standard | 94.4 | | 0.0 | | 0.0 | | 0.0 | | 0.0 | |
| + 3 epochs FT | 66.6 | -27.8 | 29.9 | 29.9 | 50.1 | 50.1 | 38.5 | 38.5 | 29.8 | 29.8 |
| + 5 epochs FT | 70.6 | -23.8 | 33.8 | 33.8 | 55.2 | 55.2 | 44.4 | 44.4 | 33.4 | 33.4 |
| + 7 epochs FT | 72.1 | -22.3 | 36.1 | 36.1 | 58.9 | 58.9 | 45.9 | 45.9 | 35.6 | 35.6 |
| + 10 epochs FT | 75.4 | -19.0 | 37.1 | 37.1 | 61.0 | 61.0 | 47.9 | 47.9 | 36.9 | 36.9 |
| + 15 epochs FT | 76.0 | -18.4 | 40.2 | 40.2 | 61.6 | 61.6 | 49.2 | 49.2 | 40.0 | 40.0 |

Table 9 shows the effect of our E-AT-fine-tuning for different numbers of epochs on either a RN-18 robust wrt $l_\infty$ or naturally trained. For the robust model, with longer training the clean accuracy progressively improves, as well as the robustness in the union of the threat models. In particular, the models fine-tuned for 15 epochs has robust accuracy similar to that achieved by the MAX-training ($43.2\%$ compared to $43.3\%$, see Table 5), while still being significantly faster even considering the training time of the initial model. When starting from a standard model, E-AT fine-tuning leads to a large drop in clean accuracy while the robustness in the union remains lower than what can be achieved with robust models, even when using 15 epochs.

Table 10: **ImageNet - Fine-tuning for more epochs:** We fine-tune the $l_2$-robust model from Engstrom et al. (2019) for either 1 or 3 epochs with our E-AT scheme.

| model | clean | | $l_\infty$ ($\epsilon_\infty = \frac{4}{255}$) | | $l_2$ ($\epsilon_2 = 2$) | | $l_1$ ($\epsilon_1 = 255$) | | union | |
|---|---|---|---|---|---|---|---|---|---|---|
| RN-50 - $l_2$ (Engstrom et al., 2019) | 58.7 | | 25.0 | | 40.5 | | 14.0 | | 13.5 | |
| + 1 epochs FT | 56.7 | -2.0 | 26.7 | 1.7 | 41.0 | 0.5 | 25.4 | 11.4 | 23.1 | 9.6 |
| + 3 epochs FT | 57.4 | -1.3 | 27.8 | 2.8 | 41.6 | 1.1 | 26.7 | 12.7 | 23.7 | 10.2 |

Additionally we fine-tune for 3 epochs, instead of 1 as done above, the $l_2$-robust model on ImageNet from Engstrom et al. (2019) with our E-AT. Table 10 shows that the longer fine-tuning improves all the performance metrics between 0.6% and 1.3%.

Table 11: **CIFAR-10 - Fine-tuning for 1 epoch:** We show the effect of fine-tuning for a single (compared to the standard 3) models robust wrt a single norm.

| model | clean | | $l_\infty$ ($\epsilon_\infty = \frac{8}{255}$) | | $l_2$ ($\epsilon_2 = 0.5$) | | $l_1$ ($\epsilon_1 = 12$) | | union | |
|---|---|---|---|---|---|---|---|---|---|---|
| RN-50 (Engstrom et al., 2019) - $l_\infty$ | 88.7 | | 50.9 | | 59.4 | | 5.0 | | 5.0 | |
| + 1 epochs FT | 84.8 | -3.9 | 46.6 | -4.3 | 68.3 | 8.9 | 47.2 | 42.2 | 42.8 | 37.8 |
| + 3 epochs FT | 86.2 | -2.5 | 46.0 | -4.9 | 70.1 | 10.7 | 49.2 | 44.2 | 43.4 | 38.4 |
| RN-50 (Engstrom et al., 2019) - $l_2$ | 91.5 | | 29.7 | | 70.3 | | 27.0 | | 23.0 | |
| + 1 epochs FT | 85.9 | -5.6 | 41.8 | 12.1 | 69.6 | -0.7 | 47.6 | 20.6 | 39.7 | 16.7 |
| + 3 epochs FT | 87.8 | -3.7 | 43.1 | 13.4 | 70.8 | 0.5 | 50.2 | 23.2 | 41.7 | 18.7 |
| RN-18 (Croce & Hein, 2021) - $l_1$ | 87.1 | | 22.0 | | 64.8 | | 60.3 | | 22.0 | |
| + 1 epochs FT | 78.9 | -8.2 | 37.7 | 15.7 | 62.9 | -1.9 | 51.3 | -9.0 | 37.6 | 15.6 |
| + 3 epochs FT | 83.5 | -3.6 | 40.3 | 18.3 | 68.1 | 3.3 | 55.7 | -4.6 | 40.1 | 18.1 |

## C.5 RESULTS ON CIFAR-10 USING A SINGLE EPOCH

Since we use a single epoch of fine-tuning on ImageNet, we here test its effect on CIFAR-10. In Table 11 we fine-tune with E-AT models adversarially trained wrt a single norm for 1 epoch: this is sufficient to significantly increase the robustness in the union of the threat models, which gets close to that obtained with the standard 3 epochs (differences are in the range 0.6% to 2.5%). In particular, the $l_\infty$-robust classifier is again the most suitable for the fine-tuning, since it has been trained in the most challenging threat model.

## C.6 FINE-TUNING WITH OTHER METHODS

Table 12: **CIFAR-10 - Other methods vs E-AT for fine-tuning:** We fine-tune with different methods for multiple norms for 3 epochs the RN-18 robust wrt individual norms (mean and standard deviation of the clean and robust accuracy over 5 seeds is reported). Additionally, we report E-AT with 6 epochs since it is at least two times faster than MAX and MSD.

| model | clean | $l_\infty$ ($\epsilon_\infty = \frac{8}{255}$) | $l_2$ ($\epsilon_2 = 0.5$) | $l_1$ ($\epsilon_1 = 12$) | union |
|---|---|---|---|---|---|
| RN-18 - $l_\infty$ - SAT | $83.5 \pm 0.23$ | $43.5 \pm 0.16$ | $68.0 \pm 0.43$ | $47.4 \pm 0.48$ | $41.0 \pm 0.26$ |
| RN-18 - $l_\infty$ - MAX | $82.2 \pm 0.33$ | $45.2 \pm 0.39$ | $67.0 \pm 0.68$ | $46.1 \pm 0.44$ | $42.2 \pm 0.56$ |
| RN-18 - $l_\infty$ - MSD | $82.2 \pm 0.42$ | $44.9 \pm 0.29$ | $67.1 \pm 0.64$ | $47.2 \pm 0.59$ | $42.6 \pm 0.17$ |
| RN-18 - $l_\infty$ - E-AT unif. | $82.6 \pm 0.62$ | $44.4 \pm 0.37$ | $68.0 \pm 0.26$ | $48.5 \pm 0.96$ | $42.2 \pm 0.34$ |
| RN-18 - $l_\infty$ - E-AT | $82.7 \pm 0.41$ | $44.3 \pm 0.63$ | $68.1 \pm 0.48$ | $48.7 \pm 0.46$ | $42.2 \pm 0.78$ |
| RN-18 - $l_\infty$ - E-AT 6 ep. | $83.2 \pm 0.41$ | $44.4 \pm 0.59$ | $68.2 \pm 0.29$ | $50.0 \pm 0.77$ | $42.3 \pm 0.64$ |
| RN-18 - $l_2$ - SAT | $86.8 \pm 0.33$ | $38.2 \pm 0.35$ | $69.6 \pm 0.69$ | $49.1 \pm 0.62$ | $37.2 \pm 0.41$ |
| RN-18 - $l_2$ - MAX | $85.1 \pm 0.83$ | $42.1 \pm 0.29$ | $69.4 \pm 0.22$ | $45.6 \pm 0.46$ | $40.0 \pm 0.33$ |
| RN-18 - $l_2$ - MSD | $85.3 \pm 0.42$ | $42.0 \pm 0.71$ | $69.2 \pm 0.23$ | $44.0 \pm 0.45$ | $39.0 \pm 0.50$ |
| RN-18 - $l_2$ - E-AT unif. | $85.9 \pm 0.51$ | $40.0 \pm 0.65$ | $69.4 \pm 0.68$ | $50.3 \pm 0.44$ | $38.9 \pm 0.79$ |
| RN-18 - $l_2$ - E-AT | $85.8 \pm 0.68$ | $40.7 \pm 0.90$ | $69.5 \pm 0.50$ | $49.5 \pm 0.54$ | $39.4 \pm 0.72$ |
| RN-18 - $l_2$ - E-AT 6 ep. | $86.3 \pm 0.22$ | $41.8 \pm 0.15$ | $70.2 \pm 0.31$ | $50.1 \pm 0.35$ | $40.5 \pm 0.31$ |
| RN-18 - $l_1$ - SAT | $85.1 \pm 0.32$ | $38.4 \pm 0.62$ | $68.3 \pm 0.40$ | $55.0 \pm 0.73$ | $38.2 \pm 0.64$ |
| RN-18 - $l_1$ - MAX | $82.2 \pm 0.34$ | $42.7 \pm 0.51$ | $66.8 \pm 0.53$ | $48.1 \pm 0.35$ | $41.5 \pm 0.38$ |
| RN-18 - $l_1$ - MSD | $81.8 \pm 0.72$ | $42.7 \pm 0.48$ | $66.8 \pm 0.36$ | $47.9 \pm 0.57$ | $41.5 \pm 0.49$ |
| RN-18 - $l_1$ - E-AT unif. | $83.5 \pm 0.68$ | $39.8 \pm 0.51$ | $68.0 \pm 0.19$ | $55.8 \pm 0.55$ | $39.6 \pm 0.47$ |
| RN-18 - $l_1$ - E-AT | $83.6 \pm 0.56$ | $40.5 \pm 0.38$ | $68.1 \pm 0.13$ | $55.3 \pm 0.38$ | $40.3 \pm 0.33$ |
| RN-18 - $l_1$ - E-AT 6 ep. | $84.2 \pm 0.36$ | $41.2 \pm 0.47$ | $68.7 \pm 0.48$ | $55.9 \pm 0.33$ | $40.9 \pm 0.57$ |

We here explore the option of fine-tuning robust models with techniques other than our E-AT, and report the results in Table 12. In particular, we test SAT, MAX and MSD for fine-tuning the RN-18 robust to single norms as done in Table 8 (the results of E-AT unif. and E-AT are taken from there).

First, one sees that SAT performs significantly worse than the other methods in this scenario, for all the fine-tuned classifiers. Second, we observe that E-AT achieves very similar results to MAX and MSD, especially when using the $l_\infty$-robust classifier which yields the highest robustness in the union. However, we highlight that MAX and MSD are 3x and 2x more expensive than E-AT (see Tables 1 and 5). Thus, we include in Table 12 E-AT with twice the budget, that is 6 epochs, which has comparable cost to MSD and closes the small gap (on average MAX and E-AT with 6 epochs perform equal and are slightly better than MSD). Since one of our goals is to reduce the cost of getting multiple-norm robust classifiers, we use E-AT as the main tool for fine-tuning.

## C.7 FINE-TUNING PERCEPTUALLY ROBUST MODELS

Table 13: **CIFAR-10 - 3 epochs of E-AT fine-tuning on $l_p$-robust models:** We use E-AT to fine-tune models robust wrt a single $l_p$-norm for multiple-norm robustness, and report the robust accuracy on 1000 test points for all threat models, and the difference compared to the initial classifier.

| model | | clean | | $l_\infty$ ($\epsilon_\infty = \frac{8}{255}$) | | $l_2$ ($\epsilon_2 = 0.5$) | | $l_1$ ($\epsilon_1 = 12$) | | union | |
|---|---|---|---|---|---|---|---|---|---|---|---|
| RN-50 - $l_\infty$ | | 88.7 | | 50.9 | | 59.4 | | 5.0 | | 5.0 | |
| (Engstrom et al., 2019) + FT | | 86.2 | -2.5 | 46.0 | -4.9 | 70.1 | 10.7 | 49.2 | 44.2 | 43.4 | 38.4 |
| RN-50 - $l_2$ | | 91.5 | | 29.7 | | 70.3 | | 27.0 | | 23.0 | |
| (Engstrom et al., 2019) + FT | | 87.8 | -3.7 | 43.1 | 13.4 | 70.8 | 0.5 | 50.2 | 23.2 | 41.7 | 18.7 |
| RN-50 - PAT | | 82.6 | | 31.1 | | 62.4 | | 33.6 | | 27.7 | |
| (Laidlaw et al., 2021) + FT | | 83.7 | 1.1 | 43.7 | 12.6 | 68.5 | 6.1 | 50.7 | 17.1 | 42.3 | 14.6 |

We test the effect of E-AT fine-tuning on a model trained to be robust to perturbations which are aligned with human perception. In particular, we use the classifier obtained with perceptual adversarial training (PAT), i.e. wrt the LPIPS metric, from Laidlaw et al. (2021), and compare it to two models with the same architecture (ResNet-50) adversarially trained wrt $l_\infty$ and $l_2$. Table 13 shows the robustness in every threat model for the original models and those obtained with 3 epochs of E-AT fine-tuning. The PAT classifier has initially the highest robustness in the union, confirming the observation of Laidlaw et al. (2021) that PAT provides some robustness to unseen attacks. After fine-tuning, all three models achieve similar worst-case robustness, with the classifier originally $l_\infty$-robust being slightly better. This shows that our E-AT fine-tuning is effective even when applied to models adversarially trained not wrt an $l_p$-norm.

# D ROBUSTNESS AGAINST UNSEEN NON $l_p$-BOUNDED ATTACKS

Table 14: **CIFAR-10 - Robustness against non $l_p$-bounded attacks:** We test the robustness of WRN-28-10 trained in different threat models against different types of attacks. Moreover, we add the PAT model from Laidlaw et al. (2021), which uses RN-50 as architecture.

| model | | clean | comm. corr. | | $l_0$ | patches | frames | fog | snow | gabor | elastic | jpeg | union | avg. |
|---|---|---|---|---|---|---|---|---|---|---|---|---|---|---|
| NAT | | 94.4 | 71.6 | | 0.1 | 8.1 | 2.6 | 47.3 | 3.9 | 35.0 | 0.2 | 0.0 | 0.0 | 12.2 |
| $l_\infty$-AT | | 81.9 | 72.6 | | 7.3 | 21.6 | 26.2 | 36.0 | 35.9 | 52.5 | 59.4 | 5.1 | 2.0 | 30.5 |
| $l_2$-AT | | 87.8 | 79.2 | | 13.2 | 25.0 | 17.7 | 44.9 | 22.1 | 43.5 | 56.6 | 14.0 | 4.5 | 29.6 |
| $l_1$-AT | | 83.5 | 75.0 | | 40.9 | 41.3 | 21.1 | 35.6 | 20.6 | 41.2 | 53.3 | 25.5 | 8.6 | 34.9 |
| PAT | | 82.6 | 76.9 | | 23.3 | 37.9 | 21.7 | 53.5 | 25.6 | 41.8 | 53.5 | 13.7 | 8.0 | 33.9 |
| E-AT | | 79.1 | 71.3 | | 39.5 | 37.7 | 30.5 | 34.8 | 33.4 | 50.2 | 58.6 | 38.7 | 15.9 | 40.4 |

We here want to study to which extent the robustness achieved with multiple norms training generalizes to unseen and possibly very different threat models on CIFAR-10. We select three sparse attacks ($l_0$-bounded, patches and frames) and five adversarial corruptions (fog, snow, Gabor noise, elastic,

$l_\infty$-JPEG) from Kang et al. (2019b). Additionally, we compute the accuracy of the classifiers on the common corruptions, which are not adversarially optimized, of CIFAR-10-C (Hendrycks & Dietterich, 2019). Table 14 shows the results against such attacks of WRN-28-10 adversarially trained wrt single norms and with E-AT, and the PAT model from Laidlaw et al. (2021), which uses RN-50 as architecture. We add also a naturally trained model as further baseline. The model trained wrt multiple norms shows much higher robustness in both the union and average of these attacks, almost 2x higher in union than the best of the other models. Moreover, for almost all threat models, the E-AT model attains the highest robustness or very close, while the best among the single $l_p$-robust models varies: this means that training for multiple norms robustness allows to summarize different kinds of robustness in the same classifier. Finally, E-AT outperforms even the PAT model which is trained wrt LPIPS and aims at generalization to unseen attacks.

**Experimental details:** To test robustness against $l_0$-attacks we use Sparse-RS (Croce et al., 2020b) with a budget of 18 pixels and 10k queries. We adopt patches of size $5 \times 5$ pixels, optimized with the PGD-based from Rao et al. (2020) (without constraints on the position of the patch on the images), and frames of width 1 pixel (Zajac et al., 2019), again optimized with PGD: in both cases we use 10 random restarts of 100 iterations. For the adversarial corruptions we use the original implementation (Kang et al., 2019b) with 100 iterations and search for a budget $\epsilon$ for which the models show different levels of robustness (in details, for fog $\epsilon = 128$, snow $\epsilon = 0.5$, Gabor noise $\epsilon = 60$, elastic $\epsilon = 0.125$, $l_\infty$-JPEG $\epsilon = 0.25$). Finally, we average the classification accuracy over the 5 severities of the common corruptions. All the statistics are on 1000 test points.

# E    EXPERIMENTS ON MNIST

Table 15: **MNIST - Comparison of full training schemes and fine-tuning with E-AT for multiple norm robustness:** We train classifier (architecture as in Maini et al. (2020)) on MNIST with different training scheme. For SAT and E-AT we report, together with the statistics over multiple random seeds, the results of the best run. Additionally, we show the results of fine-tuning the $l_2$-AT model with E-AT for different numbers of epochs, which achieves the best results. (*) AVG, MAX and MSD classifiers are those provided by Maini et al. (2020).

| model | clean | $l_\infty$ ($\epsilon_\infty = 0.3$) | $l_2$ ($\epsilon_2 = 2$) | $l_1$ ($\epsilon_1 = 10$) | union |
|---|---|---|---|---|---|
| $l_\infty$-AT | $98.9 \pm 0.12$ | $90.0 \pm 0.45$ | $8.4 \pm 1.49$ | $6.0 \pm 0.65$ | $4.2 \pm 0.73$ |
| $l_2$-AT | $98.8 \pm 0.17$ | $0.0 \pm 0.05$ | $70.7 \pm 0.26$ | $59.1 \pm 0.37$ | $0.0 \pm 0.05$ |
| $l_1$-AT | $98.8 \pm 0.09$ | $0.0 \pm 0.00$ | $45.8 \pm 0.42$ | $77.2 \pm 0.14$ | $0.0 \pm 0.00$ |
| SAT | $98.6 \pm 0.17$ | $62.0 \pm 0.86$ | $65.7 \pm 1.69$ | $61.3 \pm 1.29$ | $53.9 \pm 1.25$ |
| AVG (*) | $99.1$ | $58.6$ | $60.8$ | $22.5$ | $21.1$ |
| MAX (*) | $98.6$ | $39.4$ | $59.9$ | $25.6$ | $20.3$ |
| MSD (*) | $98.2$ | $63.7$ | $66.6$ | $51.0$ | $48.7$ |
| E-AT unif. | $98.8 \pm 0.12$ | $67.1 \pm 3.03$ | $50.2 \pm 4.93$ | $62.0 \pm 4.59$ | $45.9 \pm 4.43$ |
| E-AT | $98.7 \pm 0.12$ | $69.0 \pm 3.66$ | $56.4 \pm 3.94$ | $61.1 \pm 4.03$ | $50.6 \pm 3.89$ |
| $l_2$-AT + E-AT (3 ep.) | $96.9 \pm 0.32$ | $57.5 \pm 0.92$ | $67.8 \pm 0.58$ | $62.0 \pm 1.16$ | $54.6 \pm 0.59$ |
| $l_2$-AT + E-AT (5 ep.) | $97.4 \pm 0.21$ | $60.6 \pm 2.19$ | $65.9 \pm 0.56$ | $63.8 \pm 0.93$ | $56.2 \pm 1.16$ |

**best run**

| model | clean | $l_\infty$ ($\epsilon_\infty = 0.3$) | $l_2$ ($\epsilon_2 = 2$) | $l_1$ ($\epsilon_1 = 10$) | union |
|---|---|---|---|---|---|
| SAT | $98.8$ | $62.7$ | $67.2$ | $62.6$ | $55.3$ |
| E-AT unif. | $98.9$ | $67.4$ | $56.3$ | $63.6$ | $51.6$ |
| E-AT | $98.8$ | $71.0$ | $58.4$ | $62.0$ | $54.4$ |
| $l_2$-AT + E-AT (3 ep.) | $97.3$ | $58.0$ | $67.9$ | $62.9$ | $55.7$ |
| $l_2$-AT + E-AT (5 ep.) | $97.5$ | $61.6$ | $66.2$ | $64.0$ | $57.5$ |

We further test the different techniques on the MNIST dataset. We use the same CNN of Maini et al. (2020) as architecture and $\epsilon_\infty = 0.3$, $\epsilon_2 = 2$ and $\epsilon_1 = 10$ as threshold at which evaluating robustness, as done by Maini et al. (2020). We note that while it is an easier dataset, MNIST is challenging when it comes to adversarial training since it presents unexpected phenomena: e.g. Tramèr & Boneh (2019) noted that $l_\infty$-adversarial training induces gradient obfuscation when using

attack wrt $l_2$ and $l_1$, and both Tramèr & Boneh (2019); Maini et al. (2020) had to use many PGD-steps (up to 100), and Tramèr & Boneh (2019) even a ramp-up schedule for the $\epsilon$ during training. While other modifications to the training setup might be beneficial for some or all the methods, we just increased the number of APGD-steps to 50 for $l_1$ (see more details below). In Table 15 we compare E-AT to SAT, AVG, MAX and MSD (for the last three we use the models provided by Maini et al. (2020)). First, E-AT outperforms the available classifiers trained with AVG, MAX and MSD, meaning that even on MNIST it is a strong baseline. However, in this case, SAT, which trains on all types of perturbations, achieves better results than E-AT on average: E-AT has higher variance over runs but the best run (over multiple seeds) is close to the best one of SAT in terms of robustness in the union (55.3% vs 54.4%). Interestingly, SAT has much higher robustness wrt $l_2$ compared to E-AT, but this is somehow expected since Eq. (3) would "predict" robustness for E-AT at $\epsilon_2 \approx 1.7$ while $\epsilon_2 = 2$ is used for testing, and this is precise only for linear models. Thus the slightly worse performance of E-AT compared to SAT for the chosen radii of the threat models is to be expected from our geometric analysis.

Moreover, since we have shown that fine-tuning an $l_p$-robust model with E-AT yields high multiple norms robustness, and given that E-AT from random initialization is weak mostly wrt $l_2$, we fine-tune the $l_2$-AT classifier with E-AT. This, with just 3 or 5 epochs, significantly outperforms SAT (up to +2.3% robustness in the union), while preserving $l_2$-robustness. In total, we improve the previous SOTA for multiple-norm robustness for MNIST from 48.7% (MSD) to 57.5% (E-AT fine-tuning of an $l_2$-robust model with 5 epochs) which is a significant improvement. Note that in this case we increase the radii $\epsilon_\infty$ and $\epsilon_1$ to 0.33 and 14 respectively to preserve the $l_2$-robustness: in fact, with such values Eq. (3) yields $\epsilon_2 \approx 2.16$. However, training from random initialization, in the standard setup, with the larger thresholds leads to worse robustness in the union. We hypothesize that this is due to the increased difficulty of the task to learn: it is known that even single norm adversarial training is problematic when increasing the value of $\epsilon$ (Ding et al., 2020).

**Experimental details:** For training we use 30 epochs with cyclic learning rate (maximum value 0.05, also used for fine-tuning) and no data augmentation (other settings as for CIFAR-10). As mentioned, we use in adversarial training for multiple norms (SAT, E-AT unif. and E-AT) 50 steps of APGD for $l_1$, and to reduce the training cost we decrease to 5 those for $l_\infty$. Moreover, for training, in $l_1$-APGD we increase the parameter to control the initial sparsity of the updates to 0.1 (default is 0.05). For evaluation, we use the full AutoAttack, since on MNIST FAB (Croce & Hein, 2020a) and Square Attack (Andriushchenko et al., 2020) are at times stronger than PGD-based attacks, as shown in the original papers.

