# OpenReview forum: "Adversarial robustness against multiple $l_p$-threat models at the price of one and how to quickly fine-tune robust models to another threat model"
_ICLR.cc/2022/Conference — ICLR 2022 Submitted_

### Official Review · Reviewer_bnsW · 2021-10-25

**Correctness:** 3
**Technical Novelty And Significance:** 2
**Empirical Novelty And Significance:** 3
**Recommendation:** 6
**Confidence:** 5

**Main Review:**

### Strengths
The paper tackles an important problem of training models robust against multiple attacks efficiently. The overall paper is well written and easy to follow. The proposed evaluation compares state-of-the-art single and multi-perturbation robustness methods, and the conducted experiments on ImageNet are impressive.

---

### Weaknesses
- The paper is limited in novelty. While the fine-tuning aspect is novel, but the similarity between the $\ell_1$ and $\ell_2$ perturbation was highlighted previously in Maini et al. (2020) [1], where it was shown that the first two principal components of $\ell_1$ and $\ell_2$ adversaries are largely overlapping. Therefore, I would recommend including a discussion about their conducted analysis to clarify the contributions of the proposed method.
- Further, E-AT is simply a modification of SAT, where E-AT uses the proposed adaptive sampling instead of uniform sampling.  The gains by the proposed method are marginal in robustness and training time compared to SAT. While E-AT gains marginal improvement in union accuracy, SAT obtains higher clean accuracy. Therefore, the effectiveness of E-AT is limited from these experimental evaluations.
- The theoretical and empirical analysis is restrictive to $\ell_p$-norm.  The scalability to non-$\ell_p$ attacks (e.g., spatial attacks, patch attacks, common corruptions, unforeseen adversaries) is unclear. While $\ell_p$ attacks are a standard in the community, it is essential to show the evaluation to unseen non-$\ell_p$ attacks to demonstrate the success of the proposed method in practical scenarios.

***Other questions and comments***
- E-AT obtains lower accuracy on clean examples and across all the $\ell_p$-norms in Table 1 compared to the other baselines trained with multiple attacks. Can the authors also report the average metric following Tramèr et al. (2020) and Madaan et al. (2021) to show if E-AT effectively mitigates the robustness tradeoff between multiple attacks compared to prior works?
- How is the performance for fine-tuning on CIFAR-10 with a single epoch?
- How does the performance vary on increasing the fine-tuning epochs further in Table 8? Can the authors also report the results for ImageNet?
- The performance difference between uniform and biased sampling in E-AT is marginal (Table 5 and Table 7). Comment.
- Adding the numbers for all the bars would enhance the readability of Figure 1.

---
### Post Rebuttal

Thank you for the response and the additional experimental evaluation conducted during the rebuttal. In light of the clarifications and additional evaluations, I have increased my score to 6.

---

### References:
[1] Maini et al. (2020). Perturbation Type Categorization for Multiple  Bounded Adversarial Robustness.


**Summary Of The Paper:**

The paper tackles the problem of robustness against multiple perturbations and proposes extreme norms adversarial training (E-AT) that adaptively alternates between $\ell_1$ and $\ell_\infty$-norm. Furthermore, the paper fine-tunes Gowal et al. (2020) to improve its multi-norm robustness. Finally, the experiments are conducted on CIFAR-10 and ImageNet with APGD for training, showing the proposed method's effectiveness.

**Summary Of The Review:**

I think this paper tackles an important problem and conducts an exhaustive experimental evaluation. However, I would like to clarify the points raised in my review and would be happy to raise my score if the authors can successfully address my concerns in the discussion period.

---

> ### Author Response · Authors · 2021-11-22
> **Response to Reviewer bnsW - part 1**
>
> We thank the reviewer for the detailed comments, and reply to the main questions below.\
> \
> **”The paper is limited in novelty. While the fine-tuning aspect is novel, but the similarity between the  and  perturbation was highlighted previously in Maini et al. (2020) [1]...”**
>
> We emphasize that our proposed method is grounded on the theoretical analysis of Sec. 3.1 on the geometry of the $l_p$-balls (for $p \geq 1$, not only $p\in\{1, 2\}$) and not on the similarity among different types of perturbations as in [1]. Moreover, the observation in [1] with the PCA decomposition might be influenced by the method to generate the adversarial perturbations (PGD for both norms) but other, even black-box, attacks might yield modifications with different properties. Also, we note that [1] propose a method which goes in the opposite direction compared to ours: in fact, they want to learn to distinguish the different types of perturbations and use multiple classifiers robust wrt each of them for inference. Conversely, we suggest that a single classifier can be robust to all $l_p$-norms for $p\in[1, \infty]$ while being trained (or fine-tuned) only with $l_1$- and $l_\infty$-attacks. Finally, the method of [1] relies on adding random noise on the input, which makes it incomparable to deterministic defenses and requires adaptive attacks for evaluation (see e.g. Athalye et al. (2018)).\
> \
> **”E-AT is simply a modification of SAT, where E-AT uses the proposed adaptive sampling instead of uniform sampling. The gains by the proposed method are marginal in robustness and training time compared to SAT”**
>
> An important contribution of our paper is showing that training on all $l_p$-norm is not necessary for achieving multiple norms robustness (at SOTA level), which is in contrast to what was done by prior works including SAT. We consider this a particularly relevant insight especially when one considers the problem of unseen and non $l_p$-attacks since in such case it is not possible to use all attacks at training time. Through the experimental evaluation, we show the performance of SAT, E-AT unif. and E-AT (i.e. with the biased sampling scheme). In both case studies reported in Tables 1 and 5 E-AT performs better than SAT, and E-AT performs once better (Table 5) and once (Table 1) similarly to SAT. This shows the effectiveness of our method and the importance of the biased scheme. Moreover, Table 12 (revised version) shows that when fine-tuning for only 3 epochs models robust wrt $l_2$ or $l_1$ the biased sampling achieves better results than uniform sampling. Finally, both E-AT versions are significantly more effective than SAT when fine-tuning any $l_p$-robust model (Table 12 of the revised version).\
> \
> **”The theoretical and empirical analysis is restrictive to $\ell_p$-norm. The scalability to non- attacks (e.g., spatial attacks, patch attacks, common corruptions, unforeseen adversaries) is unclear. While  attacks are a standard in the community, it is essential to show the evaluation to unseen non- attacks to demonstrate the success of the proposed method in practical scenarios”**
>
> While the focus of our paper is on $l_p$-robustness, which is by itself challenging, we agree that the generalization to unseen different attacks is an interesting aspect to analyze. Therefore, we have added (App. D in the revised version) the results of various models (standard, trained be robust wrt individual norms, trained with E-AT) against sparse attacks ($l_0$-bounded, patches, frames), adversarial corruptions (fog, snow, Gabor noise, elastic, JPEG), and on common corruptions (CIFAR-10-C). The models robust wrt multiple norms (via E-AT in this case) shows much higher robustness in both the union and average of these attacks, almost 2x higher in union than the best of the other models. Moreover, for almost all threat models, the E-AT model attains the highest robustness or very close, while the best among the single $l_p$-robust models varies: this means that training for multiple norms robustness allow to summarize different kinds of robustness in the same classifier. Finally, since the PAT method of Laidlaw et al. (2021) aims at generalization to unseen attacks, we test their classifier in the same setup, and this achieves significantly worse robustness in the union (8.0% vs 15.9% of E-AT). We think that this is a quite interesting result showing that multiple-norm robustness generalizes well to unseen threat models.\
> \
> **”Can the authors also report the average metric…”**
>
> We have added in the appendix of the revised version (Table 6) the average robustness over the threat models for the models reported in Table 1 and in Table 5. In this metric, E-AT achieves the best results for WRN-28-10, in particular significantly outperforming MAX while having the same clean accuracy. For the smaller RN-18, E-AT unif. attains the highest average robustness, with E-AT and SAT being slightly worse.

---

> > ### Author Response · Authors · 2021-11-22
> > **Response to Reviewer bnsW - part 2**
> >
> > **”How is the performance for fine-tuning on CIFAR-10 with a single epoch?”**
> >
> > We have added such results in Table 11 (revised version) in the appendix: one epoch is sufficient to achieve robustness in the union of the threat models close to that obtained with 3 epochs. In particular, the $l_\infty$-robust classifier is again the most suitable for the fine-tuning, since it has been trained in the most challenging threat model.\
> > \
> > **”How does the performance vary on increasing the fine-tuning epochs further in Table 8?”**
> >
> > When using more than a few epochs, fine-tuning becomes similar to training from random initialization, e.g. all the models in Table 1 are trained with 30 epochs. Moreover, the learning rate schedule we use for fine-tuning, optimized for a few epochs, is likely to be not the most effective one when a budget of many epochs is available (note that in Table 9 we simply linearly rescale it according to the number of epochs).\
> > \
> > **”Can the authors also report the results for ImageNet?”**
> >
> > We fine-tune the $l_2$-robust model on ImageNet for 3 epochs, which improves all metrics (clean and robust accuracy wrt the different threat models) between 0.6% and 1.3% (see details in Table 10 in the revised version).

---

### Official Review · Reviewer_XXNy · 2021-11-01

**Correctness:** 2
**Technical Novelty And Significance:** 2
**Empirical Novelty And Significance:** 2
**Recommendation:** 3
**Confidence:** 4

**Main Review:**

Strengths:

(1) The proposed E-AT method is computationally affordable since it doesn’t use all types $\ell_{p}$ adversarial examples to do training, it only alternates between the two extreme norms.

(2) E-AT could quickly fine-tune models that are only robust against one perturbation type into models that are robust against the union of multiple $\ell_{p}$ adversarial perturbations, and the fine-tune results are shown better than simple $\ell_{p}$ norm fine-tuning.

Weaknesses:

(1) Although the paper demonstrates its motivation in Sec 3.1, the reviewer is still confused about the geometry theory proposed to defend multiple $\ell_{p}$ perturbations. It’s indeed correct for a simple one-layer linear classifier that model being robust in both $\ell_{1}$ and $\ell_{\infty}$-ball is also robust w.r.t the largest $\ell_{p}$-ball that fits into the convex hull of the union of the $\ell_{1}$ and $\ell_{\infty}$-ball. However, when it comes to the deep neural network which is in the high-dimensional manifold, there is no guarantee for the above statement since there is no such model that is absolutely robust (i.e., robust accuracy is 100%) against certain norm types on CIFAR-10 or ImageNet dataset. Being partly robust on $\ell_{1}$ and $\ell_{\infty}$ (e.g., robust accuracy is 65.88%) doesn’t mean being robust within the $\ell_{p}$-ball that fits into the convex hull of the union of the $\ell_{1}$ and $\ell_{\infty}$-ball. There should be more proof about the certified robustness of the proposed method.

(2) In Fig 1, the paper shows the model adversarial robustness against multiple norm perturbations after E-AT fine-tuning is better than L1 fine-tuning. It may be unfair since the evaluation metric for union robustness is the sample-wise worst-case accuracy, and [Ref1] demonstrates that $\ell_{1}$ and $\ell_{\infty}$ robustness are mutually exclusive so fine-tuning on $\ell_{1}$ perturbation will largely decrease the robustness on $\ell_{\infty}$ perturbation, causing the bad results under the worst-case evaluation metric. More fine-tuning is needed for the baseline methods to balance the robust accuracy on all norm types.

References

[Ref1] Tramer F, Boneh D. Adversarial Training and Robustness for Multiple Perturbations[J]. Advances in Neural Information Processing Systems, 2019, 32: 5866-5876.


**Summary Of The Paper:**

This paper mainly studies the problem of defending multiple norm adversarial perturbations. The authors propose extreme norms adversarial training (E-AT), which leverages different geometry of the $\ell_{p}$-balls to conduct adversarial training by adaptively alternating between the $\ell_{1}$ norm and $\ell_{\infty}$ norm. They also show that using E-AT fine-tune could turn $\ell_{p}$ robust model into a model that is robust against the union of $\ell_{p}$ adversarial perturbations. The authors also provide some theoretical proof for their method.

**Summary Of The Review:**

The motivation and the theoretical effect of the proposed defense should be clarified and demonstrated, and some experimental comparisons need to be improved.

---

> ### Author Response · Authors · 2021-11-22
> **Response to Reviewer XXNy**
>
> We thank the reviewer for the detailed comments. We clarify the raised concerns below. \
> \
> **”(1) Although the paper demonstrates its motivation in Sec 3.1, the reviewer is still confused about the geometry theory proposed…”**
>
> The derivation in Sec. 3.1 provides local guarantees: in fact it states that, if a classifier $f$, for a given point $x$, does not change its decision in an $l_1$-ball of radius $\epsilon_1$ and in an $l_\infty$-ball of radius $\epsilon_\infty$ centered at $x$, then we can give lower bounds on the $l_p$-robustness for $p>1$ at $x$. This means that $f$ can be robust wrt multiple norms on a subset of the test points while having average robustness over the whole test set (i.e. robust accuracy) < 100%. \
> Moreover, please note that Prop. 3.1 holds true for any possible classifier, in particular deep neural networks, whereas Th. 3.1 is specific to affine classifiers. We argue that the intuition behind using the convex hull generalizes to the architectures we use (but clearly this is just an argument and not a proof), especially since most are from the class of ReLU networks which yield piecewise affine functions, and this is supported by the good empirical results of E-AT. \
> \
> **”(2) In Fig 1, the paper shows the model adversarial robustness against multiple norm perturbations after E-AT fine-tuning is better than L1 fine-tuning. It may be unfair…”**
>
> This is a misunderstanding. Fig. 1 aims at showing two distinct results which we achieve in our paper via fine-tuning the same $l_\infty$-robust model from Gowal et al. (2020): **a)** 3 epochs of fine-tuning wrt $l_1$ gives the new SOTA $l_1$-robustness, and **b)** 3 epochs of fine-tuning with E-AT provides the new SOTA robustness wrt multiple norms. The target threat models, used for training or fine-tuning, of each classifier are highlighted in the three barplots in Fig. 1 in yellow, to distinguish it from the others (in grey) which are reported to better convey the effect of fine-tuning.\
> \
> We hope that this clarifies the motivation and the results of our method, and would be happy to integrate the Reviewer’s suggestion to make Fig. 1 clearer.

---

### Official Review · Reviewer_LQa3 · 2021-11-01

**Correctness:** 3
**Technical Novelty And Significance:** 2
**Empirical Novelty And Significance:** 2
**Recommendation:** 5
**Confidence:** 5

**Main Review:**

This paper is generally well-written. The motivation for the problem and the E-AT method are clear, and it seems like the experiments were carefully run.

Still, I have some concerns about whether the results support the proposed method. It seems like the main contribution proposed by the authors is the idea of training against only $\ell_1$ and $\ell_\infty$ attacks to get robustness to $\ell_p$ attacks for other $p$ values. However, when selecting between $\ell_1$ and $\ell_\infty$ attacks randomly, the method performs nearly identically to (or maybe a bit worse than) SAT which randomly selects between $\ell_1$, $\ell_2$, and $\ell_\infty$ attacks at each iteration. The proposed method is only superior to SAT when using the proposed biased sampling scheme. Thus, it seems like it is actually the biased sampling scheme—not the use of $\ell_1$ and $\ell_\infty$ threat models—that leads to the improvement. As far as I can tell, none of the other experiments in the paper clearly separate the effect of these two changes to SAT-type methods.

In general, the paper seems like a collection of three separate ideas:

 1. restricting adversarial training against multiple $\ell_p$ norms to only $\ell_1$ and $\ell_\infty$, which is the subject of the theory part of the paper and the motivation for the name E-AT,
 2. the biased sampling scheme,
 3. and the idea of using fine-tuning to quickly impart robustness against a different $\ell_q$ threat model to a model trained with an $\ell_p$ threat model for $q \neq p$.

As I mentioned above, the effects of 1 and 2 are currently difficult to disentangle, and 3 seems somewhat orthogonal to the other two contributions. In particular, in Table 4 it seems that adversarial fine-tuning (3) is effective even without 1 and 2. The three ideas are mostly just connected by the common problem they aim to solve—producing robustness against different and multiple $\ell_p$ norms. Even the title of the submission seems split between the different ideas. Thus, it would be helpful if the authors could better motivate the combination of E-AT and fine-tuning in the same paper, or even consider splitting the paper into two submissions.


**Summary Of The Paper:**

This paper proposes a method to produce image classifiers which are adversarially robust against multiple $\ell_p$ threat models—in particular, against $\ell_1$, $\ell_2$, and $\ell_\infty$ attacks. The method involves training against $\ell_1$ and $\ell_\infty$ attacks with the hypothesis that this will additionally give robustness for $\ell_p$ threat models with $1 \leq p \leq \infty$. This hypothesis is supported by prior results that proved that affine classifiers robust to $\ell_1$ and $\ell_\infty$ threat models are also be robust to other $\ell_p$ threat models. The authors test their method on CIFAR-10 and ImageNet for both training classifiers from scratch and for fine-tuning robust models trained on one $\ell_p$ threat model to the other $\ell_p$ threat models.

**Summary Of The Review:**

Overall, the paper could use some more work to separate the effects of the components of E-AT and motivate the combination of it with the fine-tuning contribution, which seems orthogonal. Thus, I do not recommend acceptance, although I am open to raising my score during the discussion period.

**After discussion period:** My overall position is that I would be more supportive of accepting the paper if it focused on fine-tuning over E-AT. Currently, the extensive focus on E-AT detracts from what seems like the more important contribution and could lead that contribution to be overlooked. However, I believe changing the focus would require a major revision and thus I maintain my weak reject score. Nonetheless, I respect that some of the other reviewers have different opinions on whether the paper should be published and I hope the AC weighs all our points to make the final decision.

---

> ### Author Response · Authors · 2021-11-22
> **Response to Reviewer LQa3**
>
> We thank the reviewer for the detailed comments, and address the raised concerns below.\
> \
> **”the paper seems like a collection of three separate ideas”**
>
> The common theme of the different parts of our paper is showing how it is possible to get very strong baselines for adversarial robustness in various threat models with simple and extremely efficient techniques. We first observe that doing adversarial training alternating only between the extreme norms (E-AT) is sufficient for matching (or getting very close to) the more expensive existing methods for multiple norms robustness. Based on our empirical observations, we then use E-AT to fine-tune existing models for single norm robustness, achieving with only 3 or 1 epochs of training SOTA results for CIFAR-10 and ImageNet. Finally, we study the effect of a similar short fine-tuning wrt an individual $l_p$-norm, and show that it is possible to achieve SOTA robustness in different threat models with a very limited computational cost. We think that all these contributions are very valuable for the community. The fact that only one epoch of fine-tuning is sufficient for ImageNet was quite surprising to us and is of high practical relevance.\
> \
> **”I have some concerns about whether the results support the proposed method  … separate the effects of the components of E-AT”**
>
> An important contribution of our paper is showing that training on all $l_p$-norm is not necessary for achieving multiple norms robustness (at SOTA level), which is in contrast to what was done by prior works including SAT. We consider this a particularly relevant insight especially when one considers the problem of unseen and non $l_p$-attacks since in such case it is not possible to use all attacks at training time. Through the experimental evaluation, we show the performance of SAT, E-AT unif. and E-AT (i.e. with the biased sampling scheme). In both case studies reported in Tables 1 and 5 E-AT performs better than SAT, and E-A unif. performs once better (Table 5) and once (Table 1) similarly to SAT. This shows the effectiveness of our method and the importance of the biased scheme. Moreover, Table 12 shows that when fine-tuning for only 3 epochs models robust wrt $l_2$ or $l_1$ the biased sampling achieves better results than uniform sampling, and both outperform SAT.

---

> > ### Comment · Reviewer_LQa3 · 2021-11-24
> > **Keeping my score**
> >
> > I appreciate the authors' response to my comments. However, I have decided to keep my previous score.
> >
> > While I agree that the motivation for all parts of the paper is the same, I am still not convinced that they are strongly connected enough. It seems like E-AT and adversarial finetuning are largely orthogonal, i.e. neither is required for the other to work. I would love to see the finetuning experiments given their own paper, where the authors could explore many more threat models (e.g., finetuning an $\ell_\infty$ trained model against spatial attacks) and see exactly when the finetuning works and when it doesn't. I believe a thorough empirical investigation of this would be very valuable to the community.
> >
> > I'm also still not convinced of whether the success of E-AT is due to only training against two threat models or the biased sampling scheme. I would be more convinced if the authors had included an experiment training with $\ell_1$, $\ell_2$, and $\ell_\infty$ threat models with the biased sample scheme. I am concerned that this would work just as well as E-AT, negating the need to explicitly restrict to $\ell_1$ and $\ell_\infty$ attacks. I think in fact that if the biased sample scheme is what helps, then this is more useful, because it is more transferrable to non-$\ell_p$ threat models where there is not as simple of a mathematical relationship between different threat models. For instance, I would be interested to see if one could train a model to be robust to the union of $\ell_2$, $\ell_\infty$, spatial, etc. attacks efficiently using the biased sampling scheme.

---

> > > ### Author Response · Authors · 2021-11-26
> > > **Additional response**
> > >
> > > Thanks for the additional comments.
> > >
> > > First, we note that the paper already includes a large number of experiments about fine-tuning: in fact, we explore the fine-tuning wrt single $l_p$-norms for $p\in ${$\infty, 2, 1$} and for multiple norm robustness, using pre-trained classifiers of different types (standard, robust wrt $l_p$, robust wrt perceptual metrics), architectures (from small CNNs to WideResNet-70-16), trained with and without unlabelled data, and on three datasets. Additionally, we report several ablation studies about e.g. the effect of varying the number of epochs of fine-tuning. Note that fine-tuning for multiple norms benefits from using E-AT compared to other options with the same computational cost (Table 12 of the revised version), which is introduced in the first part of the paper. We consider this a comprehensive study, which leads to SOTA results.
> > >
> > > Second, we report the results of SAT, E-AT unif. and E-AT in most of the experiments to highlight the effect of 1) not using $l_2$-attacks for training (SAT vs E-AT unif.) and 2) the biased scheme (E-AT unif. vs E-AT). Note that our intuition is not that training wrt only $l_\infty$ and $l_1$ can improve the robustness in the union, but rather that using all threat models is not necessary: this is confirmed by the fact the E-AT unif. works similarly to SAT in most of the cases. Moreover, integrating the biased scheme improves the performance, outperforming SAT especially for fine-tuning. Using the biased scheme with all threat models is an intermediate solution between SAT and E-AT.
> > >
> > > Finally, we want to highlight that the focus of our paper is on robustness wrt multiple $l_p$-norms. We have shown, in the new experiments in App. D, that multiple norm robust models improve even on a variety of unseen attacks, outperforming even the PAT model (Laidlaw et al., 2021). However, a more detailed study on such attacks and how to possibly include them during training, although certainly an interesting topic, is out of the scope of this work.

---

> > > > ### Comment · Reviewer_LQa3 · 2021-11-29
> > > > **Agree that adversarial fine-tuning is significant**
> > > >
> > > > First, I do appreciate the comprehensive experiments on fine-tuning, which I now see after looking through the appendix. I think I agree largely with Reviewer tpW1 that the material on fine-tuning is significant and useful to the community and should be published.
> > > >
> > > > My concerns are that, in the present state of the paper, the vast majority of the space is devoted to motivating and experimenting with the E-AT method. Like Reviewer tpW1, I think this method is less important than the fine-tuning results. I am still not convinced that the biased sampling scheme is not responsible for the gain in robustness. Do the authors have any results on using the biased sampling scheme with all three norms?
> > > >
> > > > My overall position is that I would be more supportive of accepting the paper if it focused on fine-tuning over E-AT. Currently, the extensive focus on E-AT detracts from what seems like the more important contribution and could lead that contribution to be overlooked. However, I believe changing the focus would require a major revision and thus I maintain my weak reject score. Nonetheless, I respect that some of the other reviewers have different opinions on whether the paper should be published and I hope the AC weighs all our points to make the final decision.

---

> > > > > ### Author Response · Authors · 2021-11-30
> > > > > **Thanks for the feedback**
> > > > >
> > > > > We appreciate the feedback. Since emphasizing the fine-tuning part is a suggestion shared by the reviewers, we will try to integrate it in the final version, by shortening the presentation of E-AT to make room for more details about fine-tuning e.g. the effect of different number of epochs and using a standard pre-trained model (currently in the appendix). At the moment we don’t have experiments about using all threat models with the biased scheme, but would be happy to add such ablation in the next version.

---

### Official Review · Reviewer_tpW1 · 2021-11-03

**Correctness:** 3
**Technical Novelty And Significance:** 2
**Empirical Novelty And Significance:** 4
**Recommendation:** 6
**Confidence:** 5

**Main Review:**

## Strengths
1. This paper makes a very strong contribution to the sub-field of multiple norm adversarial robustness by showing that one can achieve this goal by spending a small budget on any pre-trained robust model.
2. I like the graphs in Figure 4 of the Appendix because they highlight how prior methods naturally do extreme norm sampling in the given setting of CIFAR-10. However, as a word of caution here -- in my understanding, you will not see the same trend in the case of MNIST -- this makes me speculate the performance of EAT on MNIST.
3. The overall paper is written very comprehensively, and the authors have done an extensive study of various pre-trained models. The authors have already included most experiments that I would have been curious to know as a reader -- such as fine-tuning standard models, using EAT from scratch, performance on Imagenet, comparison with other prior works. I am very satisfied by the presentation.

## Suggestions/ Weaknesses
1. Can you also see the same results on MNIST? I suspect this is hard because the $\ell_\infty$ model is not robust to $\ell_2$ attacks in MNIST. I believe that CIFAR-10 is just acting as a nice test bench for this kind of work because the problem was biased against $\ell_2$ attacks (because of how perturbation budgets have been historically chosen in literature). There is no reason why the perturbation budget of interest should be 0.5, especially when you consider that the robustness of $\ell_2$ models against $\ell_2$ attacks is significantly more than that of any other $\ell_p$ robust model against $\ell_p$ attacks.
2. Comparison with CCAT or other methods such as PAT would be helpful. Can you also fine-tune models beyond $\ell_p$ robustness -- to say that this is a general property of adversarial robustness and not particularly hinged on geometry?


## Questions
1. Are you using AutoAttack (plus) or standard? I would suggest using the plus version to be comparable to other works.
2. How is EAT-unif different from AVG (if $\ell_2$ was not considered)? Noting that in most cases unif and biased sampling have nearly the same effect.
3. Do we really need to use EAT to fine-tune models? What about using MAX or MSD as a fine-tuning step? Does this perform better? Since in most results where EAT is independently used, the performance is sub-optimal. I wonder if there is any reason why EAT should be optimal for fine-tuning or if it needed to be introduced in the first place?
4. What is the impact of the additional data versus running 6 epochs of fine-tuning?

## Post-author response
I am voting for acceptance of this work. However, I also understand why the biggest contribution of this paper might be hard to grasp for the average reader. The authors spend too much time and paper space to highlight how EAT is novel. I do not think that is the case (it is a heuristic modification that works in practice!), nor do I think this paper needs that to be accepted. The remaining contribution is significant enough for me to vote for acceptance -- and in fact one of the most significant developments in multi robustness after the initial few papers. I would encourage the authors to spend more time to re-evaluate what is the most significant contribution in their opinion, and possibly rethink how to go about the prose.



**Summary Of The Paper:**

This paper addresses the problem of multiple perturbation adversarial robustness for attacks subsumed within $\ell_p$ regions for $p\in{1,2,\infty}$. The main contribution of this work is to show how a model robust to a particular attack type (typically $\ell_\infty$) can be fine-tuned (at low cost) to be robust against multiple (or alternate) perturbation types. The authors build on prior formalization about the geometry of $\ell_p$ balls (by Croce et. al.) to empirically demonstrate its effect. The results are convincing and evaluated against AutoAttack which is

**Summary Of The Review:**

The paper has a very strong contribution to the field of multi-norm adversarial robustness. However, I am concerned if this phenomenon generalizes to other datasets -- based on some observations noted by the authors. I would be happy to raise my scores if the authors can show results on MNIST as well.

---

> ### Author Response · Authors · 2021-11-22
> **Response to Reviewer tpW1 - part 1**
>
> We thank the reviewer for the detailed comments, and address the raised questions below.\
> \
> **”Can you also show the same results on MNIST?”**
>
> As asked by the Reviewer, we repeat the experiments about multiple norms robustness on MNIST (results in App. E). We note that while it is an easier dataset, MNIST is challenging when it comes to adversarial training since it presents unexpected phenomena: e.g. Tramèr & Boneh (2019) noted that $l_\infty$-adversarial training induces gradient obfuscation when using attack wrt $l_2$ and $l_1$, and both Tramèr & Boneh (2019) and Maini et al. (2020) had to use many PGD-steps (up to 100), and Tramèr & Boneh (2019) even a ramp-up schedule for the $\epsilon$ during training. We did not have time to try out such different schedules and just increased the number of APGD-steps to 50 for $l_1$. \
> In Table 15 we compare E-AT to SAT, AVG, MAX and MSD (for the last three we use the models provided by Maini et. (2020)). First, E-AT outperforms the available classifiers trained with AVG, MAX and MSD, meaning that even on MNIST it is a strong baseline. However, in this case, SAT, which trains on all types of perturbations, achieves better results than E-AT on average: E-AT has higher variance over runs but the best run (over multiple seeds) is close to the best one of SAT in terms of robustness in the union (55.3% vs 54.4%). Interestingly, SAT has much higher robustness wrt $l_2$ compared to E-AT, but this is somehow expected since Eq. (3) would “predict” robustness for E-AT at $\epsilon_2 \approx 1.7$ while $\epsilon_2=2$ is used for testing, and this is precise only for linear models. Thus the slightly worse performance of E-AT compared to SAT for the chosen radii of the threat models is to be expected from our geometric analysis. \
> Moreover, since we have shown that fine-tuning an $l_p$-robust model with E-AT yields high multiple norms robustness, and given that E-AT from random initialization is weak mostly wrt $l_2$ and as mentioned by the Reviewer, we fine-tune the $l_2$-AT classifier with E-AT (note that in this case we increase the radii $\epsilon_p$ to preserve the $l_2$-robustness, see more details in App. E). This, with just 3 or 5 epochs, significantly outperforms SAT (up to +2.3% robustness in the union), while preserving $l_2$-robustness. In total, we want to highlight that we improve the previous SOTA for multiple-norm robustness for MNIST from 48.7% (MSD) to 57.5% (E-AT fine-tuning of an $l_2$-robust model with 5 epochs).  \
> \
> **”Comparison with CCAT or other methods such as PAT would be helpful…”**
>
> Please note that CCAT is effective when using a detection threshold to filter out examples with low confidence, which makes it not comparable to our setup. Moreover, even with the thresholding, [Stutz et al. (2020)](https://arxiv.org/abs/1910.06259) report a robust accuracy of 31.6% with $\epsilon=0.03$ wrt $l_\infty$ on CIFAR-10, which is worse than the robust accuracy we achieve in the union, i.e. against stronger attacks (note that we even use $\epsilon = 8/255 > 0.03$). \
> We have added a comparison to PAT in the revised version: Table 13 shows that the original ResNet50 model of Laidlaw et al. (2021) has better robustness in the union than classifiers robust wrt an individual norm, but much lower than those trained for multiple norms. However, we fine-tune with E-AT the PAT model, and achieve comparable results to when fine-tuning $l_p$-robust models: this further shows that the fine-tuning can be applied to a variety of classifiers.\
> \
> **”Are you using AutoAttack (plus) or standard?”**
>
> As mentioned in the paragraph *“Evaluation of adversarial robustness”* we use the two versions of APGD (with cross-entropy and targeted DLR loss) as in the final version of AutoAttack (note that this includes the targeted versions of APGD and FAB which were initially only part of the *plus* version). In fact, the combination of the two APGD-attacks have been shown to provide accurate estimations of robustness on CIFAR-10 and ImageNet when simple adversarial training is used (Croce & Hein, 2020c, 2021). We tested the full AutoAttack on a subset of our models and saw no difference compared to the reported robustness. On MNIST, we use instead the full version of AutoAttack since in that case FAB and Square Attack are at times stronger than PGD-based attacks, as shown in the original papers.

---

> > ### Author Response · Authors · 2021-11-22
> > **Response to Reviewer tpW1 - part 2**
> >
> > **”How is EAT-unif different from AVG (if $\ell_2$ was not considered)? Noting that in most cases unif and biased sampling have nearly the same effect”**
> >
> > AVG (Tramèr & Boneh, 2019) runs, for each iteration, PGD-based attacks for all three threat models and then minimizes, with the training step, the average loss over them.
> > The advantage of the biased sampling scheme of E-AT can be seen in Table 1, where the robustness in the union is 1.3% better than with uniform sampling. Moreover, in Table 8 (revised version) when fine-tuning the $l_2$- and $l_1$-robust models, which are only little robust in the most challenging threat model ($l_\infty$), the biased scheme allows to focus more on $l_\infty$-robustness which improves the robustness in the union. This shows that, in particular for the more difficult cases, the biased scheme is indeed better.\
> > \
> > **”Do we really need to use EAT to fine-tune models? What about using MAX or MSD as a fine-tuning step?”**
> >
> > It is indeed possible to fine-tune with other techniques too. We have added such experiments in Table 12, where we test SAT, MAX and MSD for fine-tuning the RN-18 robust to single norms as done in Table 8 (the results of E-AT unif. and E-AT are taken from there). First, one sees that SAT performs significantly worse than the other methods in this scenario, for all the fine-tuned classifiers, which highlights the improvement achieved with E-AT and E-AT unif. by training only on the extreme norms. Second, we observe that E-AT achieves very similar results to MAX and MSD, especially when using the $l_\infty$-robust classifier which yields the highest robustness in the union. However, we emphasize that MAX and MSD are 3x and 2x more expensive than E-AT (see Tables 1 and 5). Thus, we include in Table 12 E-AT with twice the budget, that is 6 epochs, which has comparable cost to MSD and closes the small gap (on average MAX and E-AT with 6 epochs perform equal and are slightly better than MSD). Since one goal of the paper is to reduce the cost of getting multiple-norm robust classifiers, we use E-AT as the main tool for fine-tuning.\
> > \
> > **”What is the impact of the additional data versus running 6 epochs of fine-tuning?”**
> >
> > In our experiments, fine-tuning without additional data models which were trained with extra data significantly degrades the results and gets closer to that of models trained only on the standard training set.

---

> > > ### Comment · Reviewer_tpW1 · 2021-11-27
> > > **Acknowledging Author Response**
> > >
> > > Thank you for your response. I am voting for acceptance of this work. However, I also understand why the biggest contribution of this paper might be hard to grasp for the average reader. The authors spend too much time and paper space to highlight how EAT is novel. I do not think that is the case (it is a heuristic modification that works in practice!), nor do I think this paper needs that to be accepted. The remaining contribution is significant enough for me to vote for acceptance -- and in fact one of the most significant developments in multi robustness after the initial few papers. I would encourage the authors to spend more time to re-evaluate what is the most significant contribution in their opinion, and possibly rethink how to go about the prose.

---

> > > > ### Author Response · Authors · 2021-11-29
> > > > **Thanks for the feedback**
> > > >
> > > > We thank the reviewer for the support for acceptance. We appreciate the suggestion about the presentation of the contributions, and will certainly keep it in mind for improving the paper in the final version.

---

> > ### Comment · Reviewer_tpW1 · 2021-11-27
> > **Imagenet Training Details**
> >
> > Can you also provide more information about the training details on ImageNet? I see that the time for the single fine-tuning epoch is not mentioned. What is the tradeoff between the number of steps of PGD iterations and the accuracy? I am assuming this is very difficult to analyze given the computational costs? How were the decisions made? And what choices did you finalize on?

---

> > > ### Author Response · Authors · 2021-11-29
> > > **ImageNet details**
> > >
> > > For ImageNet we use 5 steps of APGD when training wrt $l_\infty$ or $l_2$, and 15 for $l_1$ (both in E-AT and for single norm fine-tuning), since optimizing in the $l_1$-ball in such high dimensional space is more challenging. Other training details are available in App. A. Then, the runtime depends on the threat model, and 1 epoch of E-AT takes around 12 hours on a single GPU. Using fewer steps slightly degrades robustness, e.g. using only 10 steps for $l_1$ in E-AT decreases $l_1$-robustness by 2-3%. However, since we use only 1 epoch of training we can afford more iterations for the generation of the adversarial points unlike with full training. We are happy to include a detailed ablation study about this aspect in the next version.

---

### Author Response · Authors · 2021-11-22
**General response**

We thank all reviewers for their helpful comments.

We are glad the reviewers appreciated the empirical results, extensive evaluation and presentation in our paper. We have revised the submission to integrate the suggestions and added the new experiments. The new parts are highlighted in blue in the text. In particular, we show (App. D) that the model trained for multiple norms robustness with E-AT is the most robust against a variety of unseen attacks, including sparse ones and adversarial corruptions. This emphasizes the relevance of a method to obtain, at small computational cost, classifiers robust wrt multiple norms, even on large datasets like ImageNet (where no such model had been achieved before).

We provide detailed answers to main points raised by the reviewers individually below.

---

### Decision · Program_Chairs · 2022-01-20

**Decision:**

Reject

**Comment:**

In this paper, authors study adversarial robustness against the union of Lp-threat models. Reviewers had some concerns about this work. They mentioned the paper is not well-organized and the explanations of the novel components should be clearer.  In particular, they suggested authors to study the effects of different components of E-AT and motivate their combinations with fine-tuning. The lack of novelty was another concern. I suggest authors to focus on the fine-tuning part in their revised draft which has more novelty. Given all, I think the paper needs a bit more work before being accepted.